# Non-backtracking Graph Neural Networks

**Seonghyun Park**[1†]**, Narae Ryu**[2†]**, Gahee Kim**[2]**, Dongyeop Woo**[1]**,**
**Se-Young Yun**[2‡]**, Sungsoo Ahn**[1‡]
*{shpark26, dongyeop.woo, sungsoo.ahn}@postech.ac.kr,    {nrryu, gaheekim, yunseyoung}@kaist.ac.kr*
[1] *POSTECH*    [2] *KAIST*

Reviewed on OpenReview: *https://openreview.net/forum?id=64HdQKnyTc*

## Abstract

The celebrated message-passing updates for graph neural networks allow representing large-scale graphs with local and computationally tractable updates. However, the updates suffer from backtracking, i.e., a message flowing through the same edge twice and revisiting the previously visited node. Since the number of message flows increases exponentially with the number of updates, the redundancy in local updates prevents the graph neural network from accurately recognizing a particular message flow relevant for downstream tasks. In this work, we propose to resolve such a redundancy issue via the non-backtracking graph neural network (NBA-GNN) that updates a message without incorporating the message from the previously visited node. We theoretically investigate how NBA-GNN alleviates the over-squashing of GNNs, and establish a connection between NBA-GNN and the impressive performance of non-backtracking updates for stochastic block model recovery. Furthermore, we empirically verify the effectiveness of our NBA-GNN on the long-range graph benchmark and transductive node classification problems.

## 1 Introduction

Recently, graph neural networks (GNNs) (Kipf & Welling, 2017; Hamilton et al., 2017; Xu et al., 2019) have shown great success in various applications, such as molecular property prediction (Gilmer et al., 2017) and community detection (Bruna & Li, 2017). Such success can be largely attributed to the message-passing structure of GNNs, which provides a computationally tractable way of incorporating the overall graph through iterative updates based on local neighborhoods. However, the message-passing structure also brings challenges due to the parallel updates and memoryless behavior of messages passed along the graph.

In particular, the message flow in a GNN is prone to backtracking, where the message from vertex $i$ to vertex $j$ is reincorporated in the subsequent message

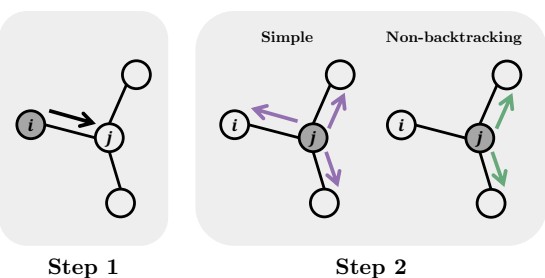

Figure 1: Comparison of two types of message flow. **Step 1**: Message flow from node $i$ to node $j$. **Step 2**: The simple update includes the message flow from node $j$ back to node $i$ (left) at step 1, while the non-backtracking update removes this redundant message flow (right).

from $j$ to $i$, e.g. the left image of step 2 in Figure 1. Since the message-passing iteratively aggregates the information, the GNN inevitably encounters an exponential surge in the number of message flows, proportionate to the vertex degrees. This issue is compounded by backtracking, which accelerates the growth of message flows with redundant information.

---

† Equal Contribution, ‡ Co-corresponding author.

The effectiveness of non-backtracking updates has been extensively explored in non-GNN message-passing algorithms or random walks (Fitzner & van der Hofstad, 2013; Rappaport et al., 2017) (Figure 1). For example, given a pair of vertices $i, j$, the belief propagation algorithm (Pearl, 1982) forbids an $i \to j$ message from incorporating the $j \to i$ message. Researchers have similarly considered non-Markovian walks (Alon et al., 2007), i.e., walks that do not consecutively traverse the same edge twice. Such classic algorithms have demonstrated great success in applications such as probabilistic graphical model inference and stochastic block models (Bordenave et al., 2015). In particular, the spectrum of the non-backtracking operator contains more useful information than that of the adjacency matrix in revealing the hidden structure of a graph model (Bordenave et al., 2015).

However, despite their promise, non-backtracking updates have received limited attention in the context of GNNs. While Chen et al. (2018) have considered combining non-backtracking updates with the typical GNN updates, the updates are not designed to prevent backtracking updates, and the analysis is limited to the optimization landscape of linear GNNs. Hence, a thorough investigation of the benefits of non-backtracking updates in GNNs is warranted.

**Contribution.** In this paper, we propose to use a non-backtracking graph neural network (NBA-GNN) to resolve the redundant messages, the over-squashing phenomenon. We employ non-backtracking updates on the messages, i.e., forbid a message from vertex $i$ to vertex $j$ from being incorporated in the message from vertex $j$ to $i$. To this end, we associate hidden features with transitions between a pair of vertices, e.g., $h_{j \to i}$, and update them from features associated with non-backtracking transitions, e.g., $h_{k \to j}$ for $k \neq i$.

To motivate our work, we formulate "message flows" as the sensitivity of a GNN concerning walks in the graph. Then, we explain how the message flows are redundant; the GNN's sensitivity of a walk with backtracking transitions is covered by other non-backtracking walks. The redundancy harms the GNN since the number of walks increases exponentially as the number of layers grows and the GNN becomes insensitive to a particular walk information. Hence, reducing the redundancy by simply considering non-backtracking walks would benefit the message-passing updates to recognize each walk's information better. As a motivating example, we provide a connection between our sensitivity analysis to the over-squashing phenomenon for GNN (Topping et al., 2022; Black et al., 2023; Di Giovanni et al., 2023) in terms of access time.

Furthermore, we analyze our NBA-GNNs from the perspective of over-squashing and their expressive capability to recover sparse stochastic block models (SBMs). To this end, we prove that NBA-GNN improves the Jacobian-based measure of over-squashing (Topping et al., 2022) compared to its original GNN counterpart.

Next, we investigate NBA-GNN's proficiency in node classification within SBMs and its ability to distinguish between graphs originating from the Erdős–Rényi model or the SBM, from the results of (Stephan & Massoulié, 2022; Bordenave et al., 2015). Unlike traditional GNNs that operate on adjacency matrices and necessitate an average degree of at least $\Omega(\log n)$, NBA-GNN demonstrates the ability to perform node classification with a substantially lower average degree bound of $\omega(1)$ and $n^{o(1)}$. Furthermore, the algorithm can accurately classify graphs even when the average degree remains constant.

Finally, we empirically evaluate our NBA-GNN on the long-range graph benchmark (Dwivedi et al., 2022) and transductive node classification problems (Sen et al., 2008; Pei et al., 2019). We observe that our NBA-GNN demonstrates competitive performance and even achieves state-of-the-art performance on the long-range graph benchmark. For the node classification tasks, we demonstrate that NBA-GNN consistently improves over its conventional GNN counterpart.

To summarize, our contributions are as follows:

- We propose NBA-GNN as a solution for the message flow redundancy problem in GNNs.

- We analyze how NBA-GNN alleviates over-squashing and is expressive enough to recover sparse stochastic block models with an average degree of $o(\log n)$.

- We empirically verify our NBA-GNN to show state-of-the-art performance on the long-range graph benchmark and consistently improve over the conventional GNNs for the transductive node classification tasks.

## 2 Related works

**Non-backtracking Algorithms.** Many classical algorithms have considered non-backtracking updates (Newman, 2013; Kempton, 2016). Belief propagation (Pearl, 1982) infers the marginal distribution on probabilistic graphical models, and has demonstrated success for tree graphs (Kim & Pearl, 1983) and graphs with large girth (Murphy et al., 1999). Moreover, Mahé et al. (2004) and Aziz et al. (2013) suggested graph kernels between labeled graphs utilizing non-backtracking walks, and Krzakala et al. (2013) first used it for node classification. Furthermore, the non-backtracking has been shown to yield better spectral separation properties, and its eigenspace contains information about the hidden structure of a graph model (Bordenave et al., 2015; Stephan & Massoulié, 2022).

**Non-backtracking and GNNs.** We also note that there have been similar approaches of applying non-backtracking to GNNs. Chen et al. (2018) first used the non-backtracking operator in GNNs, though they do not prevent backtracking and only target community detection tasks. Also, Chen et al. (2022) have computed a non-redundant tree for every node to eliminate redundancy, but inevitably suffers from high complexity. We emphasize the distinction between prior works and our NBA-GNN in Appendix A from the lens of (i) computational complexity and (ii) empirical performance.

**Over-squashing of GNNs.** When a node receives information from a $k$-hop neighbor node, an exponential number of messages pass through node representations with fixed-sized vectors. This leads to the loss of information known as *over-squashing* (Alon & Yahav, 2021), and has been formalized in terms of sensitivity (Topping et al., 2022; Di Giovanni et al., 2023). Hence, sensitivity is defined as the Jacobian of a final node feature at a GNN layer to the initial node representation and can be upper bounded via the graph topology. Stemming from this, graph rewiring methods alleviate over-squashing by adding or removing edges to compute an optimal graph (Topping et al., 2022; Black et al., 2023; Di Giovanni et al., 2023). Another line of work uses global aspects, e.g., Transformers have been applied to consider global aspects to avoid over-squashing (Ying et al., 2021; Kreuzer et al., 2021; Rampášek et al., 2022; Shirzad et al., 2023).

**Expressivity of GNNs for the SBM.** Certain studies focus on analyzing the expressive power of GNNs using variations of the SBM (Holland et al., 1983). Fountoulakis et al. (2023) established conditions for the existence of graph attention networks (GATs) that can precisely classify nodes in the contextual stochastic block model (CSBM) with high probability. Similarly, Baranwal et al. (2023) investigated the effects of graph convolutions within a network on the XOR-CSBM. The preceding works primarily focused on the probability distribution of node features, such as the distance between the means of feature vectors. On the other hand, Kanatsoulis & Ribeiro (2023) analyzed the expressivity utilizing linear algebraic tools and eigenvalue decomposition of graph operators.

## 3 Non-backtracking Graph Neural Network

### 3.1 Motivation from Sensitivity Analysis

We first explain how the conventional message-passing updates are prone to backtracking. To be specific, consider a simple, undirected graph $\mathcal{G} = (\mathcal{V}, \mathcal{E})$ and let $\mathcal{N}(i)$ denote the set of neighbor nodes of the node $i$. Each node $i \in \mathcal{V}$ is associated with a feature $x_i$. Then, the conventional graph neural networks (GNNs), i.e., message-passing neural networks (MPNNs) (Gilmer et al., 2017), iteratively update the node-wise hidden feature at the $t$-th layer $h_i^{(t)}$ as follows:

$$h_i^{(t+1)} = \phi^{(t)}\left(h_i^{(t)}, \left\{\psi^{(t)}\left(h_i^{(t)}, h_j^{(t)}\right) : j \in \mathcal{N}(i)\right\}\right), \tag{1}$$

where $\phi^{(t)}$ and $\psi^{(t)}$ are architecture-specific non-linear update and permutation invariant aggregation functions, respectively. Our key observation is that the message from a node feature $h_i^{(t)}$ to the node feature $h_j^{(t+1)}$ is reincorporated in the node feature $h_i^{(t+2)}$, e.g., Figure 3a shows the computation graph of conventional GNNs with redundant messages.

**Sensitivity Analysis.** To concretely describe the consequences of backtracking in message-passing updates, we formulate the sensitivity of the final node feature $h_i^{(T)}$ with respect to the input as follows:

$$\sum_{j \in \mathcal{V}} \frac{\partial h_i^{(T)}}{\partial h_j^{(0)}} = \sum_{s \in \mathcal{W}(i)} \prod_{t=1}^{T} \frac{\partial h_{s(t)}^{(t)}}{\partial h_{s(t-1)}^{(t-1)}}, \tag{2}$$

where $h_i^{(0)} = x_i$, $\mathcal{W}(i)$ denotes the set of $T$-step walks ending at node $i$, and $s(t)$ denotes the $t$-th node in the walk $s \in \mathcal{W}(i)$. Intuitively, this equation shows that a GNN with $T$ layers recognize the graph via aggregation of random walks with length $T$. Our key observation from Equation (2) is on how the feature $h_i^{(T)}$ is insensitive to the information from an initial node feature $h_j^{(0)}$, due to the information being "squashed" by the aggregation over the exponential number of walks $\mathcal{W}(i)$. A similar analysis has been conducted on how a node feature $h_i^{(T)}$ is insensitive to the far-away initial node feature $h_j^{(0)} = x_j$, i.e., the over-squashing phenomenon of GNNs (Topping et al., 2022).

**Redundancy of walks with backtracking.**
In particular, a walk $s$ randomly sampled from $\mathcal{W}(i)$ is likely to contain a transition that backtracks, i.e., $s(t) = s(t+2)$ for some $t < T$. The walk $s$ would be redundant since the information is contained in two other walks in $\mathcal{W}(i)$: $s(0), \ldots, s(t+1)$ and $s(0), \ldots, s(t)s(t+1), s(t+2) = s(t), s(t+3), \ldots s(T)$, as illustrated in Figure 2. This leads to the conclusion that non-backtracking walks, i.e., walks that do not contain backtracking transitions, are sufficient to

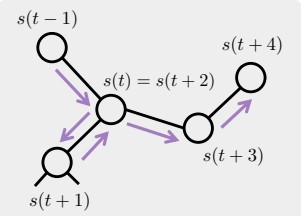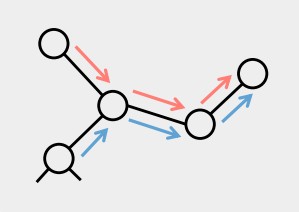

Figure 2: Two non-backtracking walks (right) are sufficient to express information contained in a walk with backtracking transition (left).

express the information in the walks $\mathcal{W}(i)$. Since the exponential number of walks in $\mathcal{W}(i)$ causes the GNN to be insensitive to a particular walk information, it makes sense to design a non-backtracking GNN that is sensitive to the constrained set of non-backtracking walks.

**Relation to Over-squashing.** Finally, we point out an intriguing motivation for our work in terms of over-squashing. Di Giovanni et al. (2023) analyzed the lower bound for the Jacobian obstruction that measures the degree of over-squashing in terms of access time with respect to a simple random walk. They concluded that the degree of over-squashing, i.e., the size of Jacobian obstruction, is higher for a pair of nodes with longer access time. Hence, for a GNN architecture robust to over-squashing, one could (i) propose a random walk that has shorter access time for a pair of nodes in the graph, and (ii) design a GNN that aligns with the random walk. Since non-backtracking random walks have been empirically shown and believed to generally yield faster access time than simple random walks (Lin & Zhang, 2019; Fasino et al., 2023), one could aim to design a GNN architecture that aligns with the non-backtracking random walks.

**Access Time of Random Walks.** However, to the best of our knowledge, there is no formal proof of scenarios where non-backtracking random walks yield a shorter access time. As a motivating example, we provide a theoretical result comparing the access time of non-backtracking and simple random walks for tree graphs. Since non-backtracking random walks do not guarantee a walk of a certain length, we make use of begrudgingly backtracking random walks (Rappaport et al., 2017), which modifies non-backtracking random walks to remove "dead ends" for tree graphs. For the full proof, please refer to Appendix B.

**Proposition 1.** *Given a tree $\mathcal{G} = (\mathcal{V}, \mathcal{E})$ and a pair of nodes $i, j \in \mathcal{V}$, the access time of begrudgingly backtracking random walk is equal to or smaller than that of a simple random walk. The equality holds if and only if the walk length is 1.*

## 3.2 Method Description

In this section, we present the **N**on-**BA**cktracking GNN (NBA-GNN) with the motivation described in Section 3.1. Given an undirected graph $\mathcal{G} = (\mathcal{V}, \mathcal{E})$, our NBA-GNN associates a pair of hidden features

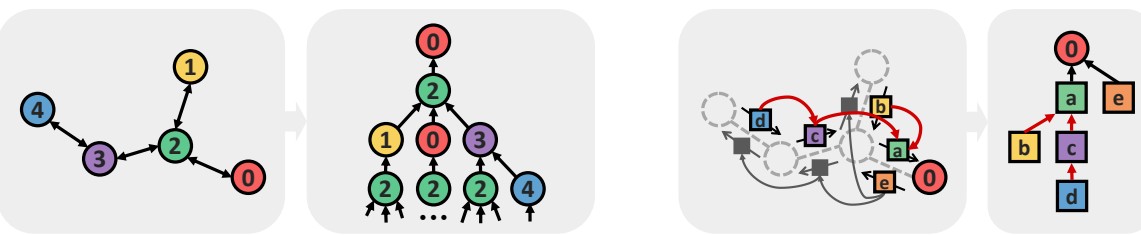

(a) Computation graph of typical GNN  (b) Computation graph of NBA-GNN

Figure 3: Computation graph of typical GNN and NBA-GNN predicting node "0". (a) Redundant messages increase the size of the computation graph, proportional to the number of layers. (b) NBA-GNN assigns a pair of features for each edge and updates them via non-backtracking message passing. By reducing redundant messages, it results in a simplified computation graph compared to typical GNNs.

$h_{i \to j}^{(t)}, h_{j \to i}^{(t)}$ for each edge $\{i, j\}$. Then the non-backtracking message passing update for a hidden feature $h_{j \to i}^{(t)}$ is defined as follows:

$$h_{j \to i}^{(t+1)} = \phi^{(t)} \left( h_{j \to i}^{(t)}, \left\{ \psi^{(t)} \left( h_{k \to j}^{(t)}, h_{j \to i}^{(t)} \right) : k \in \mathcal{N}(j) \setminus \{i\} \right\} \right), \tag{3}$$

where $\phi^{(t)}$ and $\psi^{(t)}$ are backbone-specific non-linear update and permutation-invariant aggregation functions at the $t$-th layer, respectively. For example, $\psi^{(t)}$ and $\phi^{(t)}$ are multi-layer perceptron and summation over a set for the graph isomorphism network (Xu et al., 2019, GIN), respectively. Given the update in Equation (3), one can observe that the message $h_{i \to j}^{(t)}$ is never incorporated in the message $h_{j \to i}^{(t+1)}$, and hence the update is free from backtracking. Note that line graph neural networks (Chen et al., 2018, LGNN) also applied non-backtracking operators in GNNs. However, it does not address the issue of redundant messages, as it continues to use the adjacency matrix. Our key contribution is the use of the non-backtracking operator specifically to tackle the message redundancy issue. Please refer to Appendix A for a detailed comparison.

**Initialization and Node-wise Aggregation of Messages.** The message at the 0-th layer $h_{i \to j}^{(0)}$ is initialized by encoding the node features $x_i, x_j$, and the edge feature $e_{ij}$ using a non-linear function $\phi$. After updating hidden features for each edge based on Equation (3), we apply a permutation-invariant pooling over all the messages for graph-wise predictions. Since we use hidden features for each edge, we construct the node-wise predictions at the final $T$-th layer as follows:

$$h_i = \sigma \left( \rho \left\{ h_{j \to i}^{(T)} : j \in \mathcal{N}(i) \right\}, \rho \left\{ h_{i \to j}^{(T)} : j \in \mathcal{N}(i) \right\} \right), \tag{4}$$

where $\sigma$ is a non-linear aggregation function with different weights for incoming edges $j \to i$ and outgoing edges $i \to j$, $\rho$ is a non-linear aggregation function invariant to the permutation of nodes in $\mathcal{N}(i)$. We provide a computation graph of NBA-GNN in Figure 3b to summarize our algorithm.

**Begrudgingly Backtracking Update.** While the messages from our update are resistant to backtracking, a message $h_{j \to i}^{(t)}$ may get trapped in node $i$ for the special case when $\mathcal{N}(i) = \{j\}$. To resolve this issue, we introduce a simple trick coined begrudgingly backtracking update (Rappaport et al., 2017) that updates $h_{i \to j}^{(t+1)}$ using $h_{j \to i}^{(t)}$ only when $\mathcal{N}(i) = \{j\}$. We empirically verify the effectiveness of begrudgingly backtracking updates in Section 5.3.

**Implementation.** To better understand our NBA-GNN, we provide an example of non-backtracking message-passing updates with a GCN backbone (Kipf & Welling, 2017), coined NBA-GCN. The message update at the $t$-th layer of NBA-GCN can be written as follows:

$$h_{j \to i}^{(t+1)} = h_{j \to i}^{(t)} + \sigma_{\text{GCN}} \left( \frac{1}{|\mathcal{N}(j)| - 1} \mathbf{W}^{(t)} \sum_{k \in \mathcal{N}(j) \setminus \{i\}} h_{k \to j}^{(t)} \right), \tag{5}$$

where $\sigma_{\text{GCN}}$ is an element-wise nonlinear function, e.g., rectified linear unit (Agarap, 2018, ReLU), $\mathbf{W}^{(t)}$ is the weight matrix, and messages are normalized by their number of neighbors $|\mathcal{N}(j)| - 1$.

## 4 Theoretical Analysis

In this section, we provide a theoretical analysis of the proposed NBA-GNN framework. To be specific, we show that (a) our NBA-GNNs improve the upper bound for sensitivity-based measures of GNN over-squashing and (b) NBA-GNNs can detect the underlying structure of SBMs even for very sparse graphs.

### 4.1 Sensitivity Analysis on Over-squashing

While Chen et al. (2018) initially introduced the non-backtracking operator in GNNs, they did not explore its theoretical implications. Hence, we first analyze how NBA-GNNs alleviate the over-squashing issue. A well-known quantification to assess the over-squashing effect is the *sensitivity bound* presented in Proposition 2, i.e., the Jacobian of node-wise output for another initial node feature. Note that Topping et al. (2022) assumes the node features and hidden representations as scalars for better understanding.

**Proposition 2** (**Sensitivity bounds**). *(Topping et al., 2022) Assume an MPNN defined in Equation (1). Let two nodes $i, j \in \mathcal{V}$ with distance $T$. If $\left\lVert \nabla \phi^{(t)} \right\rVert_1 \leq \alpha$ and $\left\lVert \nabla \psi^{(t)} \right\rVert_1 \leq \beta$ for $0 \leq t < T$, then the sensitivity bound can be defined as the following:*

$$\left\lVert \frac{\partial h_j^{(T)}}{\partial x_i} \right\rVert_1 \leq (\alpha\beta)^T (\widehat{A}^T)_{j,i} \,, \tag{6}$$

*where $\widehat{A}$ denotes the degree-normalized adjacency matrix.*

Over-squashing occurs when the right-hand side of Equation (6) is too small, i.e., the hidden representation of node $j$ becomes insensitive to the initial feature of node $i$ (Topping et al., 2022; Di Giovanni et al., 2023). To address this, we interpret the topology, i.e., the adjacency matrix, of the *sensitivity bound* in terms of the *random walk* that a GNN aligns with. In the following, we show that a non-backtracking random walk yields a higher sensitivity bound than simple random walks.

For our analysis, we bring the notation of non-backtracking matrix $B \in \{0,1\}^{2|\mathcal{E}| \times 2|\mathcal{E}|}$ and the incidence matrix $C \in \mathbb{R}^{2|\mathcal{E}| \times |\mathcal{V}|}$ following Chen et al. (2018) to describe the message-passing of NBA-GNNs and node-wise aggregation via linear operation, respectively. To be specific, the backtracking matrix $B$ and the incidence matrix $C$ are defined as follows:

$$B_{(\ell \to k),(j \to i)} = \begin{cases} 1 & \text{if } k = j, \ell \neq i \\ 0 & \text{otherwise} \end{cases} \quad, \quad C_{(k \to j),i} = \begin{cases} 1 & \text{if } j = i \text{ or } k = i \\ 0 & \text{otherwise} \end{cases} \quad.$$

We also define $D_{out}$ and $D_{in}$ as the matrices representing out-degree and in-degree of NBA-GNNs, respectively, capturing the count of outgoing and incoming edges for each edge. These are diagonal matrices with $(D_{out})_{(j \to i),(j \to i)} = \sum_{\ell \to k} B_{(j \to i),(\ell \to k)}$ and $(D_{in})_{(j \to i),(j \to i)} = \sum_{\ell \to k} B_{(\ell \to k),(j \to i)}$. Next, we introduce $\widehat{B}$ as the normalized non-backtracking matrix augmented with self-loops: $\widehat{B} = (D_{out} + I)^{-\frac{1}{2}}(B + I)(D_{in} + I)^{-\frac{1}{2}}$. Finally, we let $\tilde{C}$ as the matrix where $\tilde{C}_{(k \to j),i} = C_{(k \to j),i} + C_{(j \to k),i}$. Then, one obtains the following sensitivity bound of NBA-GNNs.

**Lemma 1** (**Sensitivity bounds of NBA-GNNs**). *Consider two nodes $i, j \in \mathcal{V}$ with a random walk distance $T$ given a $(T-1)$-layer NBA-GNN as described in Equation (3) and Equation (4). Suppose $\left\lVert \nabla \phi^{(t)} \right\rVert_1, \left\lVert \nabla \sigma \right\rVert_1 \leq \alpha$, $\left\lVert \nabla \psi^{(t)} \right\rVert_1, \left\lVert \nabla \rho \right\rVert_1 \leq \beta$, and $\left\lVert \nabla \phi \right\rVert_1 \leq \gamma$ for $0 \leq t < T$. Then the following holds:*

$$\left\lVert \frac{\partial h_j}{\partial x_i} \right\rVert_1 \leq (\alpha\beta)^T \gamma (\tilde{C}^\top \widehat{B}^{(T-1)} \tilde{C})_{j,i} \,.$$

We provide the proof in Appendix C. Lemma 1 states how *the over-squashing effect is controlled by the power of $\widehat{B}$*. Consequently, one can infer that increasing the upper bound likely alleviates the over-squashing

effect in GNNs (Topping et al., 2022; Black et al., 2023; Gutteridge et al., 2023) by reshaping the graph topology, i.e., the power of the adjacency matrix in the right-hand side of Equation (6). Additionally, the assumptions on nabla bounds are derived considering the Lipschitz constant of the non-linearity function and the maximum entry value across all weight matrices (Di Giovanni et al., 2023).

From this motivation, we provide an analysis to support our claim that the NBA-GNNs suffer less from the over-squashing effect due to its larger sensitivity bound. The key point is that the *sensitivity bounds align with a random walk*, and the non-backtracking random walks result in a larger sensitivity bound.

**Proposition 3.** *Consider an MPNN defined as in Equation (1) and a $(T-1)$-layer NBA-GNN described by Equation (3) and Equation (4). For any pair of nodes $i, j \in \mathcal{V}$ with distance $T$, the following inequality holds between sensitivity bounds:*

$$(\widehat{A}^T)_{j,i} \leq (\tilde{C}^\top \widehat{B}^{T-1} \tilde{C})_{j,i} \,.$$

*For d-regular graphs, $(\tilde{C}^\top \widehat{B}^{T-1} \tilde{C})_{j,i}$ decays slower by $O(d^{-T})$, while $(\widehat{A}^T)_{j,i}$ decays with $O((d+1)^{-T})$.*

We provide the full proof in Appendix C, based on comparing the degree-normalized number of non-backtracking and simple walks from node $i$ to node $j$. To the best of our knowledge, we are the first to compare the degree of over-squashing between GNNs aligned with different types of random walks. Hence, Proposition 3 indicates that NBA-GNNs has a larger sensitivity bound compared to conventional GNNs and suffers less from over-squashing, experimentally shown in Table 1 and Figures 4a and 4b. Moreover, for the sensitivity bound of $d$-regular graphs, consider the case of multiplying the power of $\widehat{B}$ or $\widehat{A}$ to a one-hot vector. Since every entry is always identical and smaller in $\widehat{B}$, all entries from the resulting vector from the non-backtracking matrix will have larger values than those from the adjacency matrix.

### 4.2 Expressive Power of NBA-GNN on SBMs

In the literature on the expressive capabilities of GNNs, comparisons with the well-known $k$-WL test are common. However, since the $k$-WL test only focuses on graph isomorphism, i.e. graph level tasks, it is inadequate for measuring the expressive power in node classification tasks. Furthermore, due to the substantial performance gap between the 1-WL (equivalent to 2-WL) and 3-WL tests, many algorithms fall into the range between these two tests, making it more difficult to compare them with each other (Huang & Villar, 2021; Wang et al., 2023). It is also worth noting that comparing GNNs with the WL test does not always accurately reflect their performance on real-world datasets.

To address these issues, several studies have turned to spectral analysis of GNNs. From a spectral viewpoint, GNNs can be seen as functions of the eigenvectors and eigenvalues of the given graph. NT & Maehara (2019) showed that GNNs operate as low-pass filters on the graph spectrum, and Balcilar et al. (2020) analyzed the use of various GNNs as filters to extract the relevant graph spectrum and measure their expressive power. Moreover, Oono & Suzuki (2020) argue that the expressive power of GNNs is influenced by the topological information contained in the graph spectrum.

The eigenvalues and the corresponding adjacency matrix eigenvectors play a pivotal role in establishing the fundamental limits of community detection in SBM, as evidenced by Yun & Proutière (2019). The adjacency matrix exhibits a spectral separation property, and an eigenvector containing information about the assignments of the vertex community becomes apparent (Lei & Rinaldo, 2015). Furthermore, by analyzing the eigenvalues of the adjacency matrix, it is feasible to determine whether a graph originates from the Erdős–Rényi (ER) model or the SBM (Erdős et al., 2013; Avrachenkov et al., 2015). However, these spectral properties are particularly salient when the average degree of the graph satisfies $\Omega(\log n)$. For graphs with average degrees $o(\log n)$, vertices with higher degrees predominate, affecting eigenvalues and complicating the discovery of the underlying structure of the graph (Benaych-Georges et al., 2019).

In contrast, the non-backtracking matrix exhibits several advantageous properties, even for constant-degree cases. In Stephan & Massoulié (2022), the non-backtracking matrix demonstrates a spectral separation property and establishes the presence of an eigenvector containing information about vertex community assignments, when the average degree only satisfies $\omega(1)$ and $n^{o(1)}$. Furthermore, Bordenave et al. (2015) have demonstrated that by inspecting the eigenvalues of the non-backtracking matrix, it is possible to discern whether a graph originates from the ER model or the SBM, even when the graph's average degree remains

Table 1: Comparison of conventional MPNNs and GNNs in the long-range graph benchmark, with and without Laplacian positional encoding (LapPE). We also denote the relative improvement by Impr.

| Model | Peptides-func | | Peptides-struct | | PascalVOC-SP | |
|---|---|---|---|---|---|---|
| | AP ↑ | Impr. | MAE ↓ | Impr. | F1 ↑ | Impr. |
| GCN | 0.5930 ± 0.0023 | | 0.3496 ± 0.0013 | | 0.1268 ± 0.0060 | |
| + NBA | 0.6951 ± 0.0024 | +17% | 0.2656 ± 0.0009 | +22% | 0.2537 ± 0.0054 | +100% |
| + NBA+LapPE | **0.7206 ± 0.0028** | +22% | **0.2472 ± 0.0008** | +29% | **0.3005 ± 0.0010** | +137% |
| GIN | 0.5498 ± 0.0079 | | 0.3547 ± 0.0045 | | 0.1265 ± 0.0076 | |
| + NBA | 0.6961 ± 0.0045 | +27% | 0.2534 ± 0.0025 | +29% | 0.3040 ± 0.0119 | +140% |
| + NBA+LapPE | **0.7071 ± 0.0067** | +29% | **0.2424 ± 0.0010** | +32% | **0.3223 ± 0.0010** | +155% |
| GatedGCN | 0.5864 ± 0.0077 | | 0.3420 ± 0.0013 | | 0.2873 ± 0.0219 | |
| + NBA | 0.6429 ± 0.0062 | +10% | 0.2539 ± 0.0011 | +26% | 0.3910 ± 0.0010 | +36% |
| + NBA+LapPE | **0.6982 ± 0.0014** | +19% | **0.2466 ± 0.0012** | +28% | **0.3969 ± 0.0027** | +38% |

constant. This capability enhances NBA-GNN's performance in both node and graph classification tasks, especially in sparse settings. These lines of reasoning lead to the formulation of the following propositions.

**Proposition 4.** *(Informal) Assume the average degree in the stochastic block model satisfies the conditions of being at least $\omega(1)$ and $n^{o(1)}$. In such a scenario, NBA-GNN can map from graph $\mathcal{G}$ to node labels.*

**Proposition 5.** *(Informal) Suppose we have a pair of graphs with a constant average degree, one generated from the stochastic block model and the other from the Erdős–Rényi model. In this scenario, NBA-GNN is capable of distinguishing between them.*

Proposition 4 suggests that even if a given graph is too sparse to extract node class information using a GNN with the adjacency matrix, NBA-GNN can still successfully classify the nodes with a probability approaching 1. Similarly, Proposition 5 extends this argument to the graph classification problem. The rationale behind these valuable properties of NBA-GNNs in sparse scenarios lies in the fact that the non-backtracking matrix $B^k$ exhibits similarity to the $k$-hop adjacency matrix, while $A^k$ is mainly influenced by high-degree vertices. This enables NBA-GNNs to extract valuable information from the spectrum of the non-backtracking matrix, aiding in the recovery of the hidden structure of the graph. For these reasons, NBA-GNNs would outperform traditional GNNs in both node and graph classification tasks, particularly in sparse graph environments. These propositions integrate prior work on the non-backtracking matrix into GNNs, contributing to a deeper understanding of the expressive power within GNN structures. Such an approach has the potential to advance the field by introducing new perspectives and methodologies within the GNN framework. For an in-depth exploration of this argument, please refer to Appendix D.

## 5 Experiment

In this section, we assess the effectiveness of NBA-GNNs across multiple benchmarks on graph classification, graph regression, and node classification tasks[1]. Additionally, we conduct a detailed comparison, including a complexity analysis, with existing methods aiming to reduce redundancy (see Appendix A). Detailed experimental information is also provided in Appendix E.

### 5.1 Long-Range Graph Benchmark

The long-range graph benchmark (Dwivedi et al., 2022, LRGB) considers a set of tasks that require learning long-range interactions. We validate our method using three datasets from the LRGB benchmark: `Peptides-func` (graph classification), `Peptides-struct` (graph regression), and `PascalVOC-SP` (node classification). We adopt performance scores from Dwivedi et al. (2022) for GNNs and from each baseline paper:

---

[1] The code is available at `https://github.com/seonghyun26/nba-gnn`

Table 2: Evaluation of NBA-GNN on LRGB. The **first-**, **second-** and **third-**best results are colored. Scores within a standard deviation of one another is considered equal. Non-reported values are denoted by -.

| Method | Model | Peptides-func AP ↑ | Peptides-struct MAE ↓ | PascalVOC-SP F1 ↑ |
|---|---|---|---|---|
| GNNs | GCN | $0.5930 \pm 0.0023$ | $0.3496 \pm 0.0013$ | $0.1268 \pm 0.0060$ |
| | GCN+LapPE | $0.6213 \pm 0.0060$ | $0.3250 \pm 0.0180$ | $0.1370 \pm 0.0077$ |
| | GIN | $0.5498 \pm 0.0079$ | $0.3547 \pm 0.0045$ | $0.1265 \pm 0.0076$ |
| | GIN+LapPE | $0.5877 \pm 0.0044$ | $0.3369 \pm 0.0026$ | $0.1302 \pm 0.0105$ |
| | GatedGCN | $0.5864 \pm 0.0077$ | $0.3420 \pm 0.0013$ | $0.2873 \pm 0.0219$ |
| | GatedGCN+LapPE | $0.6069 \pm 0.0035$ | $0.3357 \pm 0.0006$ | $0.2860 \pm 0.0085$ |
| Subgraph GNNs | MixHop-GCN | $0.6592 \pm 0.0036$ | $0.2921 \pm 0.0023$ | $0.2506 \pm 0.0133$ |
| | MixHop-GCN+LapPE | $0.6843 \pm 0.0049$ | $0.2614 \pm 0.0023$ | $0.2218 \pm 0.0174$ |
| | PathNN | $0.6816 \pm 0.0026$ | $0.2545 \pm 0.0032$ | - |
| | CIN++ | $0.6569 \pm 0.0117$ | $0.2523 \pm 0.0013$ | - |
| Transformers | Transformer+LapPE | $0.6326 \pm 0.0126$ | $0.2529 \pm 0.0016$ | $0.2694 \pm 0.0098$ |
| | GraphGPS+LapPE | $0.6535 \pm 0.0041$ | $0.2500 \pm 0.0005$ | $0.3748 \pm 0.0109$ |
| | SAN+LapPE | $0.6384 \pm 0.0121$ | $0.2683 \pm 0.0043$ | $0.3230 \pm 0.0039$ |
| | Exphormer | $0.6527 \pm 0.0043$ | $0.2481 \pm 0.0007$ | **0.3966** $\pm$ **0.0027** |
| | Graph MLP-Mixer/ViT | $0.6970 \pm 0.0080$ | **0.2449** $\pm$ **0.0016** | - |
| Graph Rewiring | DIGL+MPNN | $0.6469 \pm 0.0019$ | $0.3173 \pm 0.0007$ | $0.2824 \pm 0.0039$ |
| | DIGL+MPNN+LapPE | $0.6830 \pm 0.0026$ | $0.2616 \pm 0.0018$ | $0.2921 \pm 0.0038$ |
| | DRew-GCN+LapPE | **0.7150** $\pm$ **0.0044** | $0.2536 \pm 0.0015$ | $0.1851 \pm 0.0092$ |
| | DRew-GIN+LapPE | **0.7126** $\pm$ **0.0045** | $0.2606 \pm 0.0014$ | $0.2692 \pm 0.0059$ |
| | DRew-GatedGCN+LapPE | $0.6977 \pm 0.0026$ | $0.2539 \pm 0.0007$ | $0.3314 \pm 0.0024$ |
| State Space Models | Graph-Mamba | $0.6739 \pm 0.0087$ | **0.2478** $\pm$ **0.0016** | **0.4191** $\pm$ **0.0126** |
| **NBA-GNNs (Ours)** | NBA-GCN | $0.6951 \pm 0.0024$ | $0.2656 \pm 0.0009$ | $0.2537 \pm 0.0054$ |
| | NBA-GCN+LapPE | **0.7207** $\pm$ **0.0028** | **0.2472** $\pm$ **0.0008** | $0.3005 \pm 0.0010$ |
| | NBA-GIN | $0.6961 \pm 0.0045$ | $0.2775 \pm 0.0057$ | $0.3040 \pm 0.0119$ |
| | NBA-GIN+LapPE | **0.7071** $\pm$ **0.0067** | **0.2424** $\pm$ **0.0010** | $0.3223 \pm 0.0063$ |
| | NBA-GatedGCN | $0.6429 \pm 0.0062$ | $0.2539 \pm 0.0011$ | **0.3910** $\pm$ **0.0010** |
| | NBA-GatedGCN+LapPE | $0.6982 \pm 0.0014$ | **0.2466** $\pm$ **0.0012** | **0.3969** $\pm$ **0.0027** |

subgraph based GNNs (Abu-El-Haija et al., 2019; Michel et al., 2023; Giusti et al., 2023), graph Transformers (Kreuzer et al., 2021; Rampášek et al., 2022; Shirzad et al., 2023; He et al., 2023), graph rewiring methods (Gasteiger et al., 2019; Gutteridge et al., 2023), and state space model (Wang et al., 2024). For NBA-GNNs and NBA-GNNs with begrudgingly backtracking, we report the one with better performance. Furthermore, LapPE, i.e., Laplacian positional encoding (Dwivedi et al., 2023), is applied as it enhances the performance of NBA-GNNs in common cases.

As one can see in Table 1, NBA-GNNs show improvement regardless of the combined backbone GNNs, i.e., GCN (Kipf & Welling, 2017), GIN (Xu et al., 2019), and GatedGCN (Bresson & Laurent, 2018), aligning with our results in Proposition 3. Specifically, NBA-GCN outperforms GatedGCN, (i) confirming that simply updating both node and edge features does not lead to performance improvement and (ii) showing that non-backtracking is a key component for solving long-range interaction tasks. Furthermore, when compared to a variety of recent baselines in Table 2, at least one NBA-GNN shows competitive performance with the best baseline, except for the state space model for PascalVOC-SP. It is also noteworthy that the improvement of NBA-GNNs is higher in dense graphs, where PascalVOC-SP has an average degree of 8 while Peptides-func and Peptides-struct have an average degree of 2.

## 5.2 Transductive Node Classification Tasks

To validate the effectiveness of non-backtracking in transductive node classification tasks, we conduct experiments on three citation networks (Cora, CiteSeer, and Pubmed) (Sen et al., 2008) and three heterophilic

Table 3: Comparison of GNNs and their NBA-GNN counterpart on transductive node classification tasks for citation networks and heterophilic datasets, with and without Laplacian positional encoding (LapPE). We mark the best numbers in bold.

| Model | Cora | CiteSeer | PubMed | Texas | Wisconsin | Cornell |
|---|---|---|---|---|---|---|
| GCN | $0.8658_{\pm0.0060}$ | $0.7532_{\pm0.0134}$ | $0.8825_{\pm0.0042}$ | $0.6162_{\pm0.0634}$ | $0.6059_{\pm0.0438}$ | $0.5946_{\pm0.0662}$ |
| + LapPE | $0.8592_{\pm0.0083}$ | $0.7572_{\pm0.0132}$ | $0.8817_{\pm0.0040}$ | $0.6216_{\pm0.0584}$ | $0.6000_{\pm0.0600}$ | $0.5703_{\pm0.0547}$ |
| + NBA | $\mathbf{0.8722_{\pm0.0095}}$ | $0.7585_{\pm0.0175}$ | $0.8826_{\pm0.0044}$ | $\mathbf{0.7108_{\pm0.0796}}$ | $\mathbf{0.7471_{\pm0.0386}}$ | $0.6108_{\pm0.0614}$ |
| + NBA+LapPE | $0.8720_{\pm0.0129}$ | $\mathbf{0.7609_{\pm0.0186}}$ | $\mathbf{0.8827_{\pm0.0048}}$ | $0.6811_{\pm0.0595}$ | $\mathbf{0.7471_{\pm0.0466}}$ | $\mathbf{0.6378_{\pm0.0317}}$ |
| GraphSAGE | $0.8632_{\pm0.0158}$ | $0.7559_{\pm0.0161}$ | $0.8864_{\pm0.0030}$ | $0.7108_{\pm0.0556}$ | $0.7706_{\pm0.0403}$ | $0.6027_{\pm0.0625}$ |
| + LapPE | $0.8700_{\pm0.0117}$ | $0.7608_{\pm0.0144}$ | $\mathbf{0.8895_{\pm0.0047}}$ | $0.7162_{\pm0.0653}$ | $0.7647_{\pm0.0453}$ | $0.6189_{\pm0.0484}$ |
| + NBA | $\mathbf{0.8702_{\pm0.0083}}$ | $0.7586_{\pm0.0213}$ | $0.8871_{\pm0.0044}$ | $0.7270_{\pm0.0905}$ | $\mathbf{0.7765_{\pm0.0508}}$ | $\mathbf{0.6459_{\pm0.0691}}$ |
| + NBA+LapPE | $0.8650_{\pm0.0120}$ | $\mathbf{0.7621_{\pm0.0172}}$ | $0.8870_{\pm0.0037}$ | $\mathbf{0.7486_{\pm0.0612}}$ | $0.7647_{\pm0.0531}$ | $0.6378_{\pm0.0544}$ |
| GAT | $0.8694_{\pm0.0119}$ | $0.7463_{\pm0.0159}$ | $0.8787_{\pm0.0046}$ | $0.6054_{\pm0.0386}$ | $0.6000_{\pm0.0491}$ | $0.4757_{\pm0.0614}$ |
| + LapPE | $0.8686_{\pm0.0152}$ | $0.7512_{\pm0.0154}$ | $0.8775_{\pm0.0045}$ | $0.6135_{\pm0.0404}$ | $0.6294_{\pm0.0448}$ | $0.5108_{\pm0.0769}$ |
| + NBA | $\mathbf{0.8722_{\pm0.0120}}$ | $0.7549_{\pm0.0171}$ | $\mathbf{0.8829_{\pm0.0043}}$ | $0.6622_{\pm0.0514}$ | $0.7059_{\pm0.0562}$ | $\mathbf{0.5838_{\pm0.0558}}$ |
| + NBA+LapPE | $0.8692_{\pm0.0098}$ | $\mathbf{0.7561_{\pm0.0175}}$ | $0.8822_{\pm0.0047}$ | $\mathbf{0.6730_{\pm0.0348}}$ | $\mathbf{0.7314_{\pm0.0531}}$ | $0.5784_{\pm0.0640}$ |
| ChebNet | $0.8523_{\pm0.0110}$ | $0.7399_{\pm0.0160}$ | $0.8718_{\pm0.0029}$ | $0.6811_{\pm0.0554}$ | $0.7098_{\pm0.0322}$ | $0.6473_{\pm0.0520}$ |
| + LapPE | $0.8531_{\pm0.0139}$ | $0.7396_{\pm0.0241}$ | $0.8701_{\pm0.0047}$ | $0.6324_{\pm0.0308}$ | $0.6941_{\pm0.0905}$ | $0.6527_{\pm0.0506}$ |
| + NBA | $\mathbf{0.8823_{\pm0.0159}}$ | $0.7379_{\pm0.0099}$ | $\mathbf{0.8832_{\pm0.0062}}$ | $\mathbf{0.7568_{\pm0.0468}}$ | $\mathbf{0.7490_{\pm0.0671}}$ | $\mathbf{0.6973_{\pm0.0483}}$ |
| + NBA+LapPE | $0.8795_{\pm0.0217}$ | $\mathbf{0.7421_{\pm0.0137}}$ | $0.8821_{\pm0.0076}$ | $0.7243_{\pm0.0520}$ | $0.7451_{\pm0.0808}$ | $0.6757_{\pm0.0634}$ |

Table 4: Comparison of NBA-GNN architectures and edge representation updating GNN architectures on transductive node classification tasks. We mark the best numbers in bold.

| Model | Cora | CiteSeer | PubMed | Texas | Wisconsin | Cornell |
|---|---|---|---|---|---|---|
| NBA-GCN | $0.8722_{\pm0.0095}$ | $0.7585_{\pm0.0175}$ | $0.8826_{\pm0.0044}$ | $0.7108_{\pm0.0796}$ | $0.7471_{\pm0.0386}$ | $0.6108_{\pm0.0614}$ |
| NBA-GraphSAGE | $0.8702_{\pm0.0083}$ | $\mathbf{0.7586_{\pm0.0213}}$ | $\mathbf{0.8871_{\pm0.0044}}$ | $\mathbf{0.7270_{\pm0.0905}}$ | $\mathbf{0.7765_{\pm0.0508}}$ | $\mathbf{0.6459_{\pm0.0691}}$ |
| NBA-GAT | $0.8722_{\pm0.0120}$ | $0.7549_{\pm0.0171}$ | $0.8829_{\pm0.0043}$ | $0.6622_{\pm0.0514}$ | $0.7059_{\pm0.0562}$ | $0.5838_{\pm0.0558}$ |
| GatedGCN | $0.8477_{\pm0.0156}$ | $0.7325_{\pm0.0192}$ | $0.8671_{\pm0.0060}$ | $0.6108_{\pm0.0652}$ | $0.5824_{\pm0.0641}$ | $0.5216_{\pm0.0987}$ |
| EGNN | $\mathbf{0.8769_{\pm0.0125}}$ | $0.7567_{\pm0.0221}$ | $0.8769_{\pm0.0028}$ | $0.6595_{\pm0.0527}$ | $0.6784_{\pm0.0407}$ | $0.5946_{\pm0.0573}$ |
| CensNet | $0.8648_{\pm0.0138}$ | $0.7516_{\pm0.0162}$ | $0.8753_{\pm0.0076}$ | $0.6405_{\pm0.0510}$ | $0.6608_{\pm0.0463}$ | $0.6162_{\pm0.0707}$ |

datasets (Texas, Wisconsin, and Cornell) (Pei et al., 2019). We use three conventional GNN architectures - GCN (Kipf & Welling, 2017), GraphSAGE (Hamilton et al., 2017), and GAT (Veličković et al., 2018) - and one spectral GNN architecture, ChebNet (Defferrard et al., 2016), as the backbone of NBA-GNN in Table 3. The results indicate that the non-backtracking update improves the performance of all GNN variants. Furthermore, Table 3 shows significant enhancements in heterophilic datasets. Given that related nodes in heterophilic graphs are often widely separated (Zheng et al., 2022), the ability of NBA-GNN to alleviate over-squashing plays a vital role in classifying nodes in such scenarios.

## 5.3 Ablation Studies

In this section, we conduct ablation studies to empirically verify our framework. For simplicity, we use BA for backtracking GNNs and BG for begrudgingly backtracking GNNs. All experiments are averaged over 3 seeds, and hyper-parameters for GCN from Tönshoff et al. (2023) are used for a fair comparison.

**Non-backtracking vs. Backtracking.** We first verify whether the performance improvements indeed stem from the non-backtracking updates. To this end, we compare our NBA-GNN with a backtracking variant, coined BA-GNN. To be clear, BA-GNN allows backtracking update prohibited in NBA-GNN, i.e., using $h_{i \to j}^{(\ell)}$ to update $h_{j \to i}^{(\ell+1)}$ in Equation (3). From Figures 4a and 4b, NBA-GCN consistently outperforms the BA-GCN and GCN regardless of the number of layers, aligning with the results of Proposition 3. Intriguingly, one can also observe that BA-GCN outperforms the naïve backbone, i.e., GCN, consistently.

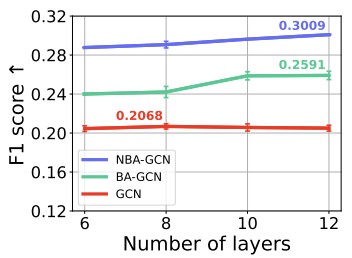

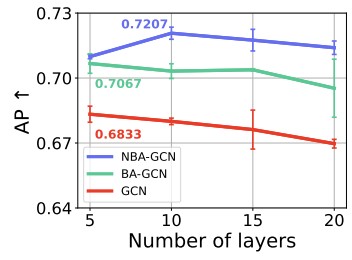

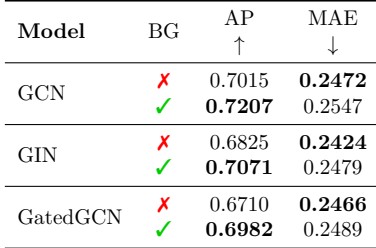

(a) F1 for changes in number of layers in `PascalVOC-SP`

(b) AP for changes in number of layers in `Peptides-func`

(c) Performance of begrudgingly, non-backtracking in sparse graphs

Figure 4: Ablation studies on the components of NBA-GNN.

Table 5: Training time and memory usage on the large datasets for the transductive node classification task.

| Datasets | Model | Time (s) | # Param. (K) | Memory (MiB) | Model | Time (s) | # Param. (K) | Memory (MiB) |
|---|---|---|---|---|---|---|---|---|
| Cora | GCN | 12.22 | 2317 | 24.01 | GCN+NBA | 38.95 | 3051 | 158.86 |
| | SAGE | 10.74 | 3104 | 27.01 | SAGE+NBA | 47.07 | 3837 | 161.86 |
| | GAT | 17.72 | 3130 | 27.21 | GAT+NBA | 58.20 | 3864 | 162.06 |
| CiteSeer | GCN | 15.74 | 3479 | 60.68 | GCN+NBA | 52.44 | 5375 | 373.36 |
| | SAGE | 14.95 | 4265 | 63.68 | SAGE+NBA | 59.94 | 6161 | 376.36 |
| | GAT | 19.68 | 4292 | 63.87 | GAT+NBA | 66.92 | 6188 | 376.56 |
| PubMed | GCN | 42.66 | 1838 | 47.9 | GCN+NBA | 242.66 | 2094 | 436.9 |
| | SAGE | 56.10 | 2624 | 50.9 | SAGE+NBA | 320.92 | 2880 | 439.9 |
| | GAT | 64.80 | 2650 | 51.0 | GAT+NBA | 416.85 | 2906 | 440.0 |

**Begrudgingly Backtracking Updates in Sparse Graphs.** Additionally in Figure 4c, we investigate the effect of begrudgingly backtracking in sparse graphs, i.e., `Peptides-func`, and `Peptides-struct`. One can see that begrudgingly backtracking is effective in `Peptides-func`, and shows similar performance in `Peptides-struct` (Note that the `PascalVOC-SP` does not have a vertex with degree one).

**Non-backtracking & Edge Features Updates.** From another point of view, NBA-GNN can be interpreted as an architecture only updating edge-wise features using backbone GNN layers. Therefore, we conduct additional experiments to validate that the performance improvement truly comes from non-backtracking rather than updating edge features in Table 3. We consider three GNN architectures that also update edge features: GatedGCN (Bresson & Laurent, 2018), EGNN (Gong & Cheng, 2019), and CensNet (Jiang et al., 2020). As seen in Table 4, NBA-GNN outperforms these models on most datasets, showing the effectiveness of non-backtracking over edge feature updates. It is noteworthy that NBA-GNN only updates edge features derived from initial node features, while EGNN and CensNet update both node and edge features from initial node and edge features. We report detailed implementation in Appendix E.3.3.

## 5.4 Complexity analysis

In this section, we analyze the space and time complexity of some baselines and NBA-GNN. All experiments were conducted on a single RTX 3090.

**Space Complexity.** NBA-GNNs generate messages for each edge considering directions and pass these messages in a non-backtracking matter. This process requires $2|\mathcal{E}|$ messages, and $(d_{avg} - 1)|\mathcal{E}|$ connections among messages where $d_{avg}$ is the average degree of the graph. Although this may seem substantial, it has not been a bottleneck in practice and can be mitigated by adjusting the batch size. Moreover, this is a relatively less computation compared to DRew, which requires computation over $k$-hop neighbors (with $k$ being the number of layers). We report the memory usage for the transductive node classification task in Table 5 and the LRGB task in Table 6 (DRew and PathNN could not fit into a single GPU due to

Table 6: Average test time (s) per epoch and memory usage for `Peptides-func`, `Peptides-struct`, `PascalVOC-SP` dataset. Parenthesis refers to the batch size, and the best performance for each dataset is highlighted in bold.

| Metric | Dataset | PathNN-SP | PathNN-SP+ | PathNN-AP | GraphGPS+LapPE | DRew-GCN+LapPE | NBA-GCN+LapPE |
|---|---|---|---|---|---|---|---|
| **Memory usage** | `Peptides-func` | OOM (128) | OOM (128) | OOM (128) | 89.70% (128) | 46.82% (128) | **18.86% (128)** |
| | `Peptides-struct` | OOM (128) | OOM (128) | OOM (128) | 88.08% (128) | 46.82% (128) | **16.98% (128)** |
| | `PascalVOC-SP` | N/A | N/A | N/A | **45.64% (32)** | OOM (32) | 47.30% (32) |
| **Test time (s)** | `Peptides-func` | 3.349 (64) | 5.156 (64) | 5.977 (64) | 0.639 (128) | 0.933 (128) | **0.541 (200)** |
| | `Peptides-struct` | 2.120 (64) | 1.417 (64) | 1.372 (64) | 0.926 (128) | 1.053 (128) | **0.532 (128)** |
| | `PascalVOC-SP` | N/A | N/A | N/A | **1.307 (32)** | 6.468 (20) | 5.866 (30) |

Table 7: Comparison on theoretical and practical time complexity for preprocessing `Peptides-func` dataset.

| **Model** | RFGNN | PathNN-SP | DRew | NBA-GNN |
|---|---|---|---|---|
| Theoretical time complexity | $|\mathcal{V}|!/|\mathcal{V}-k-1|!$ | $|\mathcal{V}| * b^k$ | $\sum_{i=1}^{k} i * |\mathcal{E}|$ | $d_{avg} * |\mathcal{E}|$ |
| Practical computation time (s) | N/A | 666 | 123 | **68** |

memory overflow for `PascalVOC-SP`). NBA-GNN shows less memory usage compared to graph Transformer architectures, although it consumes more memory than MPNNs.

**Time Complexity.** We also report the average test time per epoch for LRGB in Table 6. Every experiment has been conducted with the largest batch size that fits the GPU. NBA-GNN shows competitive time compared to other baselines, though it does suffer in graphs with a high average degree.

**Preprocessing Time Complexity.** Finally, we investigate the time complexity of preprocessing in RFGNN (Chen et al., 2022), PathNN-SP (Michel et al., 2023), DRew (Gutteridge et al., 2023), and NBA-GNN on the `Peptides-func` dataset. To be specific, RFGNN requires $k$-depth non-redundant tree, PathNN-SP needs the single shortest path between nodes, and DRew demands the $k$-hop neighbors information. NBA-GNN requires the computation of the non-backtracking edge adjacency, which can be computed in $\mathcal{O}(|\mathcal{E}|^2)$, even $\mathcal{O}(d_{avg}|\mathcal{E}|)$ if the data is provided in the form of an adjacency list. We denote $b$ as the branching factor, $k$ as the number of layers, and $d_{avg}$ as the average degree of nodes in a graph. Inevitably, previous works are dependent on the number of layers, while NBA-GNN remains **irrelevant to the number of layers**. In Table 7, one can see that NBA-GNN shows superiority in both theoretical time complexity and practical computation time (RFGNN codes were not reported by the authors).

## 6 Conclusion

We have introduced a message-passing framework applicable to any GNN architectures to alleviate over-squashing. As theoretically shown, NBA-GNNs mitigate over-squashing in terms of sensitivity and enhance their expressive power for both node and graph classification tasks on SBMs. Additionally, we have demonstrated that NBA-GNNs achieve competitive performance on the LRGB benchmark, and show improvements over conventional GNNs across transductive node classification tasks, even in heterophilic datasets.

## Acknowledgements

S. Park and S. Ahn were supported by Institute of Information & communications Technology Planning & Evaluation (IITP) grant funded by the Korea government (MSIT) (No. RS-2019-II191906, Artificial Intelligence Graduate School Program(POSTECH)), the National Research Foundation of Korea (NRF) grant funded by the Korea government (MSIT) (No. 2022R1C1C1013366), and Basic Science Research Program through the National Research Foundation of Korea (NRF) funded by the Ministry of Education(2022R1A6A1A0305295413). N. Ryu and S. Yun were supported by Institute of Information & communications Technology Planning & Evaluation (IITP) grant funded by the Korea government (MSIT)

(No.2019-0-00075, Artificial Intelligence Graduate School Program (KAIST), 10%) and the Institute of Information & communications Technology Planning & Evaluation (IITP) grant funded by the Korea government (MSIT) (No. 2022-0-00871, Development of AI Autonomy and Knowledge Enhancement for AI Agent Collaboration, 90%).

We thank Soo Yong Lee, Seongsu Kim, Hyosoon Jang, Yunhui Jang, Juwon Hwang, Hyomin Kim, for their valuable comments and suggestions in preparing the early version of the manuscript.

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

# Appendix

# A   Comparison with Related Works

This section delves into the detailed exploration of key related works, highlighting essential distinctions from our NBA-GNN. In addition, we present simple experiments to prove the superiority of our model.

## A.1   Line Graph Neural Networks

Line Graph Neural Networks (LGNN) (Chen et al., 2018) were the pioneers in applying the non-backtracking operator to GNNs. However, it does not resolve redundancy issues, as it employs both an adjacency matrix and non-backtracking matrix in a layer. In other words, our main contribution over LGNN is in consideration of the non-backtracking operator specifically for the message redundancy problem, in a end-to-end manner. Notably, the computational complexity of LGNN is acknowledged to be close to $|V| \log(|V|)$ by its authors.

Table 8: Comparison of LGNN and NBA-GCN on `Peptides-func`.

| Model | Training AP ↑ | Test AP ↑ | Test time per epoch (s) ↓ | GPU usage (%) ↓ |
|---|---|---|---|---|
| LGNN | 0.4202 | 0.3778 | 87.761 | 97.10 |
| NBA-GCN+LapPE | **0.9724** | **0.7207** | **0.541** | **51.18** |

In our experiments on `Peptides-func`, we compare the performance and complexity of LGNN and NBA-GCN+LapPE, as detailed in Table 8. Given that LGNN was initially proposed using only node degrees for node features, we incorporated an additional node feature encoder, similar to our NBA-GNN setup. Specifically, we used a batch size of 64, hidden dimension of 24, and 4 layers for LGNN. Refer to Appendix E.2 for the hyperparameters of NBA-GCN. The drawbacks of LGNN in terms of time and space complexity become evident when compared to NBA-GNN, while requiring space and time for computing both simple random walks and non-backtracking walks, but result in poor performance.

## A.2   Redundancy-free Graph Neural Network

Redundancy-Free Graph Neural Network (RFGNN) (Chen et al., 2022) shares the common motivation with NBA-GNN, aiming to reduce redundant messages in the computation graph. RFGNN achieves this by constructing a tree for every node, coined Truncated ePath Tree (TPT). TPT ensures that there are no repeated nodes along the simple path from the root to the leaf, except for the root node which can appear twice in the path. This approach differs significantly from NBA-GNN, which eliminates redundancy by identifying non-backtracking edge adjacency.

In terms of complexity, RFGNN is known to have a space and time complexity of $\mathcal{O}\left(\frac{|V|!}{|V-t-1|!}\right)$, where $|V|$ is the number of nodes and $t$ is the number of GNN layers. In contrast, NBA-GNN achieves superior complexity with space complexity $\mathcal{O}(2|\mathcal{E}|)$ and time complexity $\mathcal{O}(d_{avg}|\mathcal{E}|)$, which remains **irrelevant to the number of layers**. The preprocessing process time complexity, involved in finding non-backtracking edge adjacency, is $\mathcal{O}(|\mathcal{E}|^2)$. This scalability allows NBA-GNN to handle larger graph dataset compared to RFGNN. Furthermore, in theoretical aspects, our work distinguishes itself by providing the sensitivity upper bound of non-backtracking updates, rather than comparing the relative inference of paths.

Table 9: Average test accuracy of LGNN, NBA-GNN and BP on two sparse SBMs, where $n$ represents the number of vertices, $C$ is the number of classes, and $p$ and $q$ denote edge probabilities within and across communities, respectively.

| Model | Parameters $(n, C, p, q)$ | LGNN | NBA-GNN | BP |
|---|---|---|---|---|
| (a) Binary assortative SBM | $(400, 2, 20/n, 10/n)$ | 0.4885 | 0.4900 | 0.5303 |
| (b) 5-community dissociative SBM | $(400, 5, 0, 18/n)$ | 0.1821 | 0.1888 | 0.2869 |

### A.3    Stochastic Block Models

We also conduct a comparative analysis for our method on two sparse Stochastic Block Models (SBMs) with distinct parameters for Belief propagation (BP), and LGNN: (a) a binary assortative SBM ($n = 400$, $C = 2$, $p = 20/n$, $q = 10/n$), and (b) a 5-community dissociative SBM ($n = 400$, $C = 5$, $p = 0$, q = $18/n$). To be specific, $n$ denotes the number of vertices, $C$ denotes the number of classes, and $p$ and $q$ denote edge probabilities within and across communities, respectively. Table 9 illustrates the average test accuracy across 100 graphs. Notably, BP, known for achieving the information-theoretic threshold, exhibited the best performance, consistent with expectations. Additionally, NBA-GNN outperforms LGNN in both scenarios.

# B  Proofs for Section 3.1

In this section, we present the findings discussed in Section 3.1. Similar to Di Giovanni et al. (2023), we concentrate on the relationship between over-squashing and access time of non-backtracking random walks. Our study establishes that the access time using a begrudgingly backtracking random walk (BBRW) is smaller than that of a simple random walk (SRW) between two nodes, in tree graphs. Also, it is noteworthy that the gap between these two access times increases as the length of the walk grows.

In the following, we will compare the access times between BBRW and SRW. First, we show that on the tree graph, the access time equals the sum of access times between neighboring nodes. Note that the access time between neighboring nodes, which is the cut-point, can be represented in terms of return time. The formulas for return time in Fasino et al. (2023) and Lemma 6 allow us to derive the formulas for access time between neighboring nodes. Finally, we derive and compare the access time of BBRW and SRW, i.e., Proposition 1.

**Proposition 1.** *Given a tree $\mathcal{G} = (\mathcal{V}, \mathcal{E})$ and a pair of nodes $i, j \in \mathcal{V}$, the access time of begrudgingly backtracking random walk is equal to or smaller than that of a simple random walk. The equality holds if and only if the walk length is 1.*

## B.1  Preliminaries

### B.1.1  Simple and Begrudgingly Random Walks

A random walk on a graph $\mathcal{G} = (\mathcal{V}, \mathcal{E})$ is a sequence of $\mathcal{V}$-valued random variable $x_0, x_1, x_2 \ldots$ where $x_{n+1}$ is chosen randomly from neighborhood of $x_n$. Different types of random walks have different probabilities for selecting neighboring nodes.

Simple random walk (SRW) choose a next node $j$ uniformly from the neighbors of current node $i$:

$$P(x_{n+1} = j | x_n = i) = \begin{cases} \frac{1}{d_i} & \text{if } (i,j) \in \mathcal{E} \\ 0 & \text{otherwise} \end{cases}.$$

On the other hand, begrudgingly backtracking random walk (BBRW) tries to avoid the previous node when there is another option:

$$P(x_{n+2} = k | x_{n+1} = j, x_n = i) = \begin{cases} \frac{1}{d_j - 1} & \text{when } (j,k) \in \mathcal{E}, \ k \neq i, \text{ and } |\mathcal{N}(j) \setminus \{i\}| \geq 1 \\ 1 & \text{when } (j,k) \in \mathcal{E} \text{ and } |\mathcal{N}(j) \setminus \{i\}| = 0 \\ 0 & \text{otherwise} \end{cases}.$$

### B.1.2  Access Time

Consider a SRW starting at a node $i \in \mathcal{V}$. Let $T_i$ denote the time when a SRW first arrived at node $i$, $T_i := \min\{n \geq 0 | x_n = i\}$, and $\overline{T}_i$ be the time when a random walk first arrived at node $i$ after the first step, $\overline{T}_i := \min\{n > 0 | x_n = i\}$. With slight abuse of notation, we define access time $t(i,j)$ from $i$ to $j$, access time $t(i \to j, k)$ from $i$ to $k$ where $x_1 = j$, and return time $t(i; \mathcal{G})$ in a graph $\mathcal{G}$ as follows:

$$t(i,j) := \mathbb{E}[T_j | x_0 = i]$$
$$t(i \to j, k) := \mathbb{E}[T_k | x_1 = j, x_0 = i]$$
$$t(i; \mathcal{G}) := \mathbb{E}[\overline{T}_i | x_0 = i]$$

Similarly, we denote the access time from $i$ to $j$, access time from $i$ to $k$ with where $x_1 = j$ and return time of BBRW as $\tilde{t}(i,j), \tilde{t}(i \to j, k)$ and $\tilde{t}(i; \mathcal{G})$, respectively. Note that the first step of BBRW shows the same behavior as the SRW since there is no previous node on the first step.

Finally, for a tree graph $\mathcal{G}$ and pair of nodes $i, j$ in Proposition 1, we denote the unique paths between $i$ and $j$ as $(v_0, \ldots, v_N)$ with $i = v_0$, $j = v_N$. We also let $N$ denote the distance between the nodes $i$ and $j$.

### B.2 Access Time of Simple Random Walks

In this section, we derive the access time of simple random walks (SRW) on tree graphs between two nodes $i$ and $j$, i.e., $\mathtt{t}(i, j)$. To be specific, we show this in the following process.

1. We decompose the access time for two nodes $i$ and $j$ into a summation of the access time of neighboring nodes in the path, i.e., $\mathtt{t}(v_n, v_{n+1})$ for $n \in 0, \cdots, N-1$ (Appendix B.2.1 and Lemma 2).

2. We evaluate the access time of neighboring nodes in the path, e.g., $\mathtt{t}(v_n, v_{n+1})$, using the number of edges in a graph (Appendix B.2.2 and Lemma 3).

3. We formulate the access time of simple random walks (SRW) between two nodes $i$ and $j$, i.e., $\mathtt{t}(i, j)$ (Appendix B.2.3 and Proposition 6).

#### B.2.1 Decomposition of Access Time

First, we show that the access time is equal to the sum of access times between neighboring nodes. When a random walker travels from $v_0$ to $v_N$, it must pass through all nodes $v_n$ on the paths. Intuitively, We can consider the entire time taken as the summation of the time intervals between when the walker arrived at $v_n$ and when it arrived at $v_{n+1}$.

**Lemma 2.** *Given a tree $\mathcal{G}$ and path $(v_0, \ldots, v_N)$,*

$$\mathtt{t}(v_0, v_N) = \sum_{n=0}^{N-1} \mathtt{t}(v_n, v_{n+1}) \,.$$

*Proof.* Given a unique path $(v_0, \ldots, v_N)$ between $v_0$ and $v_N$, any walk between $(v_0, v_N)$ can be decomposed into a series of $N-1$ walks between $(v_0, v_1), (v_1, v_2), \ldots (v_{N-1}, v_N)$ such that the walk between $(v_n, v_{n+1})$ does not contain a node $v_{n+1}$ except at the end point. The expected length of each walk is $t(v_n, v_{n+1})$.

To be specific, consider the following decomposition:

$$\mathtt{t}(v_0, v_N) = \mathbb{E}[T_{v_N} | \mathrm{x}_0 = v_0] = \sum_{n=0}^{N-1} \mathbb{E}[T_{v_{n+1}} - T_{v_n} | \mathrm{x}_0 = v_0] \,. \tag{7}$$

From the Markov property of SRW:

$$\mathbb{E}[T_{v_{n+1}} - T_{v_n} | \mathrm{x}_0 = v_0] = \mathtt{t}(v_n, v_{n+1}) \,.$$

Finally, we can formulate the access time as the following:

$$\mathtt{t}(v_0, v_N) = \sum_{n=0}^{N-1} \mathbb{E}[T_{v_{n+1}} - T_{v_n} | \mathrm{x}_0 = v_0] = \sum_{n=0}^{N-1} \mathtt{t}(v_n, v_{n+1}) \,.$$

$\square$

#### B.2.2 Access Time between Neighbors

Now, we derive the formula for the access time between neighboring nodes.

**Lemma 3.** *Given a tree graph $\mathcal{G}$ and adjacent nodes $i, j$, the associated access time $\mathtt{t}(i, j)$ is defined as follows:*

$$\mathtt{t}(i, j) = 1 + 2|\mathcal{E}(\mathcal{G}_i)| \,,$$

*where $\mathcal{E}(\mathcal{G})$ is edge set of graph $\mathcal{G}$ and $\mathcal{G}_i$ is the subtree produced by deleting edge $(i, j)$ and choosing connected component of $i$.*

*Proof.* Given a random walk from $i$ to $j$ that only contains $j$ once, every time the walk lands at the node $i$, the next step can be categorized into two scenarios:

1. The walk transitions from node $i$ to $j$ based on transitioning with respect to the edge $(i,j) \in \mathcal{E}$ with probability $\frac{1}{d_i}$. The walk terminates.

2. The walk fails to reach $j$ and continues the walk in the subtree $\mathcal{G}_i$ until arriving at the node $i$ again. Note that the walk cannot arrive at node $j$ without arriving at node $i$ in prior. In other words, the walk continues for the return time of $i$ concerning the graph $\mathcal{G}_i$.

The two scenarios imply that every time the walk arrives at node $i$, the walk terminates with probability $\frac{1}{d_i}$. Then the number of trials follows the geometric distribution and consequently, the average number of trials is $d_i$. In other words, the walk falls into the scenario of type 2, for $d_i - 1$ times on average. We have to traverse at least one edge $(i,j)$. The expected total penalty is the product of the average failure penalty and the average number of failures. Thus,

$$\mathtt{t}(i,j) = 1 + (d_i - 1)\mathtt{t}(i;\mathcal{G}_i), \tag{8}$$

where $\mathtt{t}(i;\mathcal{G}_i)$ is the return time of $i$ with respect to the subgraph $\mathcal{G}_i$. Since the return time is the ratio of the sum of the degree to the degree of the node, as stated by Fasino et al. (2023), the following equation holds:

$$\mathtt{t}(i;\mathcal{G}_i) = \frac{2|\mathcal{E}(\mathcal{G}_i)|}{d_i - 1},$$

which directly implies our conclusion of $\mathtt{t}(i,j) = 1 + 2|\mathcal{E}(\mathcal{G}_i)|$. □

### B.2.3 Main Result

Using Lemma 2 and Lemma 3, we arrive at our main result for the access time of SRW on trees.

**Proposition 6.** *Given a tree $\mathcal{G}$ and pair of nodes $i, j \in \mathcal{V}$, the following equations hold for the access time of SRW between $u$ and $v$.*

$$\mathtt{t}(i,j) = \sum_{n=0}^{N-1} \left(1 + 2|\mathcal{E}(\mathcal{G}_n)|\right),$$

*where $\mathcal{E}(\mathcal{G})$ is edge set of graph $\mathcal{G}$ and $\mathcal{G}_n$ is the subtree produced by deleting edge $(v_n, v_{n+1})$ and choosing connected component of $v_n$.*

*Proof.* The proof is a straightforward combination of Lemma 2 and Lemma 3.

$$\mathtt{t}(i,j) = \sum_{n=0}^{N-1} \mathtt{t}(v_n, v_{n+1}) = \sum_{n=0}^{N-1} \left(1 + 2|\mathcal{E}(\mathcal{G}_n)|\right).$$

□

### B.3 Access Time of Begrudgingly Backtracking Random Walks

In this section, we derive the access time of begrudgingly backtracking random walks BBRW) between two nodes $i$ and $j$, i.e., $\tilde{\mathtt{t}}(i,j)$, on tree graphs. At a high level, this section consists of the following order.

1. We decompose the access time for two nodes $v_0$ and $v_N$ into summations of the access time of neighboring nodes in the path (Appendix B.3.1 and Lemma 4).

2. The decomposed term in 1. can be formulated using (i) the return time and (ii) the access time between neighboring nodes (Appendix B.3.2 and Lemma 5).

3. The return time of a node can be formulated in terms of the number of edges and the degree of a node (Appendix B.3.3 and Lemma 6).

4. The access time between neighboring nodes can be formulated using the number of edges and the degree of the node (Appendix B.3.4 and Lemma 7).

5. Finally we formulate the access time of begrudgingly backtracking random walks (BBRW), i.e., $\tilde{\mathtt{t}}(i,j)$ (Appendix B.3.5 and Proposition 7).

#### B.3.1 Decomposition of Access Time

We start by decomposing the access time $\tilde{\mathtt{t}}(i,j)$ similar to the one in Lemma 2. However, a key difference exists in the BBRW random walk. When the walk first arrives at node $v_n$, it previously passed the node $v_{n-1}$. Therefore, we cannot return to $v_{n-1}$ upon the first failure to reach $v_{n+1}$.

**Lemma 4.** *Consider a tree graph $\mathcal{G}$ and path $(v_0, \ldots, v_N)$. Then the access time of BBRW between node $v_0$ and $v_N$ can be derived as follows:*

$$\tilde{\mathtt{t}}(v_0, v_N) = \tilde{\mathtt{t}}(v_0 \to v_1, v_0) + \sum_{n=1}^{N-1} \left( \tilde{\mathtt{t}}(v_{n-1} \to v_n, v_{n+1}) - 1 \right).$$

*Proof.* The proof is similar to Lemma 2 such that we decompose the walk into a series of $N-1$ walks between $(v_0, v_1), (v_1, v_2), \ldots, (v_{N-1}, v_N)$ such that the walk between $v_n, v_{n+1}$ does not contain a node $v_{n+1}$ except at the endpoint. The expected length of each walk is $\tilde{\mathtt{t}}(v_{n-1} \to v_n, v_{n+1}) - 1$. Note that $\tilde{\mathtt{t}}(v_{n-1} \to v_n, v_{n+1})$ counts the length of walk from $v_{n-1}$ to $v_{n+1}$, hence should be substracted to represent the length of walk from $v_n$ to $v_{n+1}$.

To be specific, we have noted that Equation (7) holds for general random walks:

$$\tilde{\mathtt{t}}(v_0, v_N) = \sum_{n=0}^{N-1} \mathbb{E}[T_{v_{n+1}} - T_{v_n} | \mathrm{x}_0 = v_0].$$

Next, when the BBRW random walk reaches $v_n$ at $T_{v_n}$, it has passed node $v_{n-1}$, i.e., $\mathrm{x}_{T_{v_n}-1} = v_{n-1}$. Therefore, the random walk after the time $t = T_{v_n} - 1$ is the random walk starting at $v_{n-1}$ with the second node being $\mathrm{x}_1 = v_n$. Therefore,

$$\mathbb{E}[T_{v_{n+1}} - T_{v_n} | \mathrm{x}_0 = v_0] = \tilde{\mathtt{t}}(v_{n-1} \to v_n, v_{n+1}) - 1 \quad \text{for} \quad n \geq 1.$$

$\square$

#### B.3.2 Expressing Transition-conditioned Access Time Using Subgraph-return Time

Next, we formulate the transition-conditioned access time $\tilde{\mathtt{t}}(v_{n-1} \to v_n, v_{n+1})$.

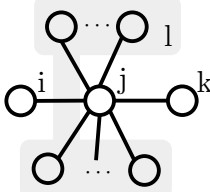

(a) Figure of nodes $i, j, k, l$ in B.3.2

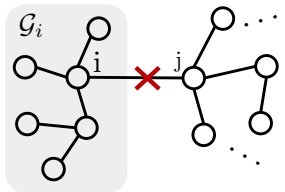

(b) Subtree $\mathcal{G}_i$ produced by deleting edge (i, j) and choosing connected components to node $i$

Figure 5: References for section B.3.2

**Lemma 5.** *Given a tree $\mathcal{G} = (\mathcal{V}, \mathcal{E})$ and three nodes $i$, $j$, and $k$ such that $(i,j), (j,k) \in \mathcal{E}$,*

$$\tilde{\mathtt{t}}(i \to j, k) - 1 = \tilde{\mathtt{t}}(j, k) - \frac{d_i - 1}{d_j}\tilde{\mathtt{t}}(i; \mathcal{G}_i) - \frac{2}{d_j}, \tag{9}$$

*where $\tilde{\mathtt{t}}(i; \mathcal{H})$ is return time of node $i$ in a subgraph $\mathcal{H}$, and $\mathcal{G}_i$ is the subtree produced by deleting edge $(i,j)$ and choosing connected components of $i$.*

*Proof.* We start with establishing basic equalities for access time of BBRW.

The first equality describes how the access time of a random walk from $i \to j$ to $k$ can be expressed by its subset $j \to l, k$ where $l \neq i$ due to the non-backtracking property,

$$\tilde{\mathtt{t}}(i \to j, k) - 1 = \frac{1}{d_j - 1} + \sum_{l \in \mathcal{N}(j) \setminus \{i, k\}} \frac{1}{d_j - 1}\tilde{\mathtt{t}}(j \to l, k). \tag{10}$$

The next equality describes how the access time from $j$ to $k$ can be decomposed by considering subsequent transitions to some $l$ in the neighborhood of $j$.

$$\tilde{\mathtt{t}}(j, k) = \frac{1}{d_j} \sum_{l \in \mathcal{N}(j)} \tilde{\mathtt{t}}(j \to l, k). \tag{11}$$

When plugging in Equation (11) into Equation (10), one can see that we need to consider $l = i, k$ for $\tilde{\mathtt{t}}(j \to l, k)$. In the case of $k$, it is trivially 1. For the case of $i$, i.e., $\tilde{\mathtt{t}}(j \to i, k)$, it can be defined as follow:

$$\tilde{\mathtt{t}}(j \to i, k) = 1 + (d_i - 1)\tilde{\mathtt{t}}(i; \mathcal{G}_i) + \tilde{\mathtt{t}}(i \to j, k). \tag{12}$$

This is similar to Equation (8), where $\mathcal{G}_i$ describes how the walk from $j \to i$ to $k$ can be divided into two scenarios: (i) continues the walk in $\mathcal{G}_i$ or (ii) $i$ transitions into $j$ with probability $1/(d_i - 1)$.

Starting from Equation (10), one can derive the following relationship:

$$
\begin{aligned}
\tilde{\mathtt{t}}(i \to j, k) - 1 &= \frac{1}{d_j - 1} \cdot 1 + \sum_{l \in \mathcal{N}(j) \setminus \{i, k\}} \frac{1}{d_j - 1}\tilde{\mathtt{t}}(j \to l, k) \\
&\overset{(a)}{=} \frac{1}{d_j - 1} + \frac{1}{d_j - 1}\left[ \sum_{l \in \mathcal{N}(j)} \tilde{\mathtt{t}}(j \to l, k) - \tilde{\mathtt{t}}(j \to i, k) - \tilde{\mathtt{t}}(j \to k, k) \right] \\
&\overset{(b)}{=} \frac{1}{d_j - 1} + \frac{1}{d_j - 1}\left[ d_j\tilde{\mathtt{t}}(j, k) - \tilde{\mathtt{t}}(j \to i, k) - 1 \right] \\
&\overset{(c)}{=} \frac{1}{d_j - 1}\left[ d_j\tilde{\mathtt{t}}(j, k) - (d_i - 1)\tilde{\mathtt{t}}(i; \mathcal{G}_i) - \tilde{\mathtt{t}}(i \to j, k) - 1 \right],
\end{aligned}
$$

where $(a)$ is from Equation (10), $(b)$ is from Equation (11) and $\tilde{\mathtt{t}}(j \to k, k) = 1$, $(c)$ is from Equation (12).
Now, by multiplying $d_j - 1$ on both sides, we get

$$(d_j - 1)\left(\tilde{\mathtt{t}}(i \to j, k) - 1\right) = d_j\tilde{\mathtt{t}}(j, k) - (d_i - 1)\tilde{\mathtt{t}}(i; \mathcal{G}_i) - \tilde{\mathtt{t}}(i \to j, k) - 1$$

$$= d_j\tilde{\mathtt{t}}(j, k) - (d_i - 1)\tilde{\mathtt{t}}(i; \mathcal{G}_i) - \left(\tilde{\mathtt{t}}(i \to j, k) - 1\right) - 2 \,.$$

Thus, we can conclude for $\tilde{\mathtt{t}}(i \to j, k) - 1$:

$$\tilde{\mathtt{t}}(i \to j, k) - 1 = \tilde{\mathtt{t}}(j, k) - \frac{d_i - 1}{d_j}\tilde{\mathtt{t}}(i; \mathcal{G}_i) - \frac{2}{d_j} \,.$$

$\square$

### B.3.3 Return Time with respect to a Subgraph

Now, we formulate the return time $\tilde{\mathtt{t}}(i; \mathcal{G})$ for a tree-graph $\mathcal{G}$. Considering the return time, we prove that the return time of BBRW is less than or equal to that of SRW.

**Lemma 6.** *Given a tree $\mathcal{G} = (\mathcal{V}, \mathcal{E})$ and a node $i$, the return time of $i$ for a BBRW is the following:*

$$\tilde{\mathtt{t}}(i; \mathcal{G}) = \frac{2|\mathcal{E}|}{d_i} \,.$$

*Proof.* Consider the tree as a rooted tree where the root is $i$. In the following, we will use mathematical induction based on the tree height of $i$.

First, consider the base case where the height of the tree is 1. Then, whatever we choose as the next node $\mathtt{x}_1$ from $\mathtt{x}_0 = i$, we return to $i$ at the second transition (i.e., $\mathtt{x}_2 = i$) since all the neighbors of $i$ are leaf node. Since $d_i = |\mathcal{E}|$ for tree with height 1, $\tilde{\mathtt{t}}(i; \mathcal{G}) = 2 = \frac{2|\mathcal{E}|}{d_i}$.

Now, assume that the lemma holds for the tree with a height less than $k \geq 1$. It suffices to show that the lemma also holds for a tree with height $k + 1$ and its root $i$. From the same perspective in the proofs of the Lemma 3, we can view the random walk returning to $i$ as follows:

1. We choose a node $\mathtt{x}_1 = l$ from $\mathcal{N}(i)$ uniformly. Then, from node $l$, we return to $l$ to reach $i$. (i.e., we have to pay penalty amounts to return time of $l$)

2. In node $l$, we try to reach $i$ by selecting edge $(l, i)$ with probability $\frac{1}{d_l - 1}$. Thus, the average number of failures is $d_l - 2$.

3. If we fail to reach $i$ from $l$, we should return to $l$ for a next chance to reach $i$. i.e., an average failure penalty amounts to a return time of $l$ in the subtree with root $l$.

Hence,

$$\tilde{\mathtt{t}}(i; \mathcal{G}) = 1 + \sum_{l \in \mathcal{N}(i)} \frac{1}{d_i}\left\{1 + \tilde{\mathtt{t}}(l; \mathcal{G}_l) \cdot \left((d_l - 2) + 1\right)\right\} = 2 + \sum_{l \in \mathcal{N}(i)} \frac{1}{d_i} \cdot \tilde{\mathtt{t}}(l; \mathcal{G}_l) \cdot (d_l - 1) \,,$$

where $\tilde{\mathtt{t}}(l; \mathcal{G}_l)$ is the return time from node $l$ for graph $\mathcal{G}$ and $\mathcal{G}_l$ denotes the subtree with root $l$, respectively.

Since the subtree with root $l$ has height less than $k$, $\tilde{\mathtt{t}}(l;\mathcal{G}_l) = \frac{2|\mathcal{E}(G_l)|}{d_l-1}$. Therefore,

$$
\begin{aligned}
\tilde{\mathtt{t}}(i;\mathcal{G}) &= 2 + \frac{1}{d_i} \sum_{l \in \mathcal{N}(i)} \tilde{\mathtt{t}}(l;\mathcal{G}_l) \cdot (d_l - 1) \\
&= 2 + \frac{1}{d_i} \sum_{l \in \mathcal{N}(i)} \frac{2|\mathcal{E}(\mathcal{G}_l)|}{d_l - 1} \cdot (d_l - 1) \\
&= 2 + \frac{1}{d_i} \sum_{l \in \mathcal{N}(i)} 2|\mathcal{E}(\mathcal{G}_l)| \\
&= \frac{2\left(d_i + \sum_{l \in \mathcal{N}(i)} |\mathcal{E}(\mathcal{G}_l)|\right)}{d_i} \\
&= \frac{2|\mathcal{E}|}{d_i} .
\end{aligned}
$$

$\square$

### B.3.4  Access Time between Neighbors

For access time between neighbors, we achieve a similar result to Lemma 3.

**Lemma 7.** *Given a tree $\mathcal{G}$ and adjacent nodes $i, j$,*

$$
\tilde{\mathtt{t}}(i,j) = 1 + 2|\mathcal{E}(\mathcal{G}_i)| \cdot \frac{d_i - 1}{d_i},
$$

*where $\mathcal{E}(\mathcal{G})$ is edge set of graph $\mathcal{G}$, and $\mathcal{G}_i$ is the subtree produced by deleting edge $(i,j)$ and choosing connected component of $i$.*

*Proof.* We follow the same perspective of the proofs in Lemma 3.

We first analyze the success probability for each trial, which is $\frac{1}{d_i}$ since there is no previous node. After the first trial, the success probability is fixed to $\frac{1}{d_i-1}$. On each trial, the average failure penalty amounts to $\tilde{\mathtt{t}}(i;\mathcal{G}_i)$, which is the return time of $i$ on subtree $\mathcal{G}_i$.

Thus, the average number of trials until first success is,

$$
(\text{Average number of trial}) = \frac{1}{d_i} \cdot 1 + \frac{d_i - 1}{d_i} \cdot \left(1 + (d_i - 1)\right).
$$

The average number of failures is as follows:

$$
\begin{aligned}
(\text{Average number of failure}) &= (\text{Average number of trial}) - 1 \\
&= \frac{1}{d_i} \cdot 1 + \frac{d_i - 1}{d_i} \cdot (1 + d_i - 1) - 1 \\
&= \frac{(d_i - 1)^2}{d_i} .
\end{aligned}
$$

Thus, the expected total penalty is as follows:

$$
(\text{Expected total penalty}) = \tilde{\mathtt{t}}(i;\mathcal{G}_i) \cdot \frac{(d_i - 1)^2}{d_i} .
$$

Since $\tilde{\mathtt{t}}(i;\mathcal{G}_i) = \frac{2|\mathcal{E}(\mathcal{G}_i)|}{d_i-1}$ by Lemma 6, $\tilde{\mathtt{t}}(u,v) = 1 + 2|\mathcal{E}(\mathcal{G}_i)|\frac{d_i-1}{d_i}$. $\square$

### B.3.5 Main Result

Finally, we show that the access time of BBRW between two nodes $i$ and $j$, i.e., $\tilde{\mathtt{t}}(i,j)$, in Proposition 7.

**Proposition 7.** *Given a tree $\mathcal{G}$ and a pair of nodes $i,j$, the following equations hold for the access time of BBRW between $i$ and $j$.*

$$\tilde{\mathtt{t}}(i,j) = \sum_{n=0}^{N-1}\left(1 + 2|\mathcal{E}(\mathcal{G}_n)| \cdot \frac{d_{v_n}-1}{d_{v_n}}\right) - \sum_{n=1}^{N-1}\frac{2|\mathcal{E}(\mathcal{G}_{n-1})|}{d_{v_n}} - \sum_{n=1}^{N-1}\frac{2}{d_{v_n}},$$

*where $\mathcal{E}(\mathcal{G})$ is the edge set of $\mathcal{G}$ and $\mathcal{G}_n$ is the subtree produced by deleting edge $(v_n, v_{n+1})$ and choosing connected component of $v_n$.*

*Proof.* By Lemma 4 and Equation (9) of Lemma 5,

$$\begin{aligned}
\tilde{\mathtt{t}}(v_0, v_N) &= \tilde{\mathtt{t}}(v_0, v_1) + \sum_{n=1}^{N-1}\left\{\tilde{\mathtt{t}}(v_{n-1} \to v_n, v_{n+1}) - 1\right\} \\
&= \tilde{\mathtt{t}}(v_0, v_1) + \sum_{n=1}^{N-1}\left\{\tilde{\mathtt{t}}(v_n, v_{n+1}) - \frac{d_{v_{n-1}}-1}{d_{v_n}}\tilde{\mathtt{t}}(v_{n-1}) - \frac{2}{d_{v_n}}\right\} \\
&= \sum_{n=0}^{N-1}\tilde{\mathtt{t}}(v_n, v_{n+1}) - \sum_{n=1}^{N-1}\frac{d_{v_{n-1}}-1}{d_{v_n}}\tilde{\mathtt{t}}(v_{n-1}) - \sum_{n=1}^{N-1}\frac{2}{d_{v_n}} \\
&= \sum_{n=0}^{N-1}\left(1 + 2|\mathcal{E}(\mathcal{G}_l)| \cdot \frac{d_{v_n}-1}{d_{v_n}}\right) - \sum_{n=1}^{N-1}\frac{2|\mathcal{E}(\mathcal{G}_{l-1})|}{d_{v_n}} - \sum_{n=1}^{N-1}\frac{2}{d_{v_n}}.
\end{aligned}$$

$\square$

### B.4   Proof of Proposition 1

Finally, we compare the access time of two random walks in a tree. Recall Proposition 1.

**Proposition 1.** *Given a tree $\mathcal{G} = (\mathcal{V}, \mathcal{E})$ and a pair of nodes $i, j \in \mathcal{V}$, the access time of begrudgingly backtracking random walk is equal to or smaller than that of a simple random walk. The equality holds if and only if the walk length is 1.*

*Proof.* Let $\mathcal{E}(\mathcal{G})$ be the edge set of $\mathcal{G}$ and $\mathcal{G}_n$ be the subtree produced by deleting edge $(v_n, v_{n+1})$ and choosing connected component of $v_n$. Then,

$$
\begin{aligned}
\tilde{\mathtt{t}}(v_0, v_N) - \mathtt{t}(v_0, v_N) &= \sum_{n=0}^{N-1} \left( 1 + 2|\mathcal{E}(\mathcal{G}_n)| \cdot \frac{d_{v_n} - 1}{d_{v_n}} \right) - \sum_{n=1}^{N-1} \frac{2|\mathcal{E}(\mathcal{G}_{n-1})|}{d_{v_n}} \\
&\quad - \sum_{n=1}^{N-1} \frac{2}{d_{v_n}} - \sum_{n=0}^{N-1} (1 + 2|\mathcal{E}(\mathcal{G}_n)|) \\
&= - \sum_{n=0}^{N-1} \frac{2|\mathcal{E}(\mathcal{G}_n)|}{d_{v_n}} - \sum_{n=1}^{N-1} \frac{2|\mathcal{E}(\mathcal{G}_{n-1})|}{d_{v_n}} - \sum_{n=1}^{N-1} \frac{2}{d_{v_n}} \leq 0 \,.
\end{aligned}
$$

Therefore the access time of two nodes is always less or equal in begrudgingly backtracking random walks than simple random walks, where equality holds for random walks with length 1.

$\square$

# C Proofs for Section 4.1

## C.1 Preliminaries

Let $\mathcal{G}$ be a graph with a set of $n$ vertices $\mathcal{V}$, a set of m edges $\mathcal{E} \in \mathcal{V}^2$. We use $x_i$ to denote the node-wise feature for $i \in \mathcal{V}$, and $d_i$ for the degree of node $i \in \mathcal{V}$. The adjacency matrix $A \in \mathbb{R}^{n \times n}$ encodes the connectivity of graph $\mathcal{G}$. For node $i \in \mathcal{V}$, we define the set of incident outgoing edges of $i$ as $N_e^+(i)$, the set of incident incoming edges of $i$ as $N_e^-(i)$, and let $N_e(i) = N_e^+(i) \cup N_e^-(i)$ be the set of all incident edges of node $i$. Also, recall the non-backtracking matrix $B \in \{0,1\}^{2|\mathcal{E}| \times 2|\mathcal{E}|}$ and the incidence matrix $C \in \mathbb{R}^{2|\mathcal{E}| \times |\mathcal{V}|}$:

$$B_{(\ell \to k),(j \to i)} = \begin{cases} 1 & \text{if } k = j, \ell \neq i \\ 0 & \text{otherwise} \end{cases} \quad , \quad C_{(k \to j),i} = \begin{cases} 1 & \text{if } j = i \text{ or } k = i \\ 0 & \text{otherwise} \end{cases} \quad .$$

We also define $D_{out}$ and $D_{in}$ as the out-degree and in-degree matrices of NBA-GNN, respectively, counting the number of outgoing and incoming edges for each edge. These are diagonal matrices with $(D_{out})_{(j \to i),(j \to i)} = \sum_{\ell \to k} B_{(j \to i),(\ell \to k)}$ and $(D_{in})_{(j \to i),(j \to i)} = \sum_{\ell \to k} B_{(\ell \to k),(j \to i)}$, for each index $j \to i$. Next, we introduce $\widehat{B}$ as the normalized non-backtracking matrix augmented with self-loops: $\widehat{B} = (D_{out} + I)^{-\frac{1}{2}}(B + I)(D_{in} + I)^{-\frac{1}{2}}$. Then one obtains the following sensitivity bound of NBA-GNN. We also let $\tilde{C}$ denote a matrix where $\tilde{C}_{(k \to j),i} = C_{(k \to j),i} + C_{(j \to k),i}$.

Consequently, one can express our NBA-GNN updates:

$$h_{j \to i}^{(t+1)} = \phi^{(t)} \left( h_{j \to i}^{(t)}, \sum_{(k,\ell) \in \mathcal{E}} \widehat{B}_{(\ell \to k),(j \to i)} \psi^{(t)} \left( h_{\ell \to k}^{(t)}, h_{j \to i}^{(t)} \right) \right), \tag{13}$$

where $\phi^{(t)}$ and $\psi^{(t)}$ corresponds to the update and the aggregation as described in Equation (3). Next, the node-wise aggregation step can be described as follows:

$$h_i = \sigma \left( \sum_{(j,k) \in \mathcal{E}} C_{(k \to j),i} \rho \left( h_{k \to j}^{(T)} \right), \sum_{(j,k) \in \mathcal{E}} C_{(j \to k),i} \rho \left( h_{j \to k}^{(T)} \right) \right).$$

## C.2 Proof of Lemma 1

The original sensitivity bound from Topping et al. (2022) for a hidden representation of node $j$ and an initial feature of node $i$ is defined as the following:

**Proposition 2** (**Sensitivity bounds**). *(Topping et al., 2022) Assume an MPNN defined in Equation (1). Let two nodes $i, j \in \mathcal{V}$ with distance $T$. If $\left\| \nabla \phi^{(t)} \right\|_1 \leq \alpha$ and $\left\| \nabla \psi^{(t)} \right\|_1 \leq \beta$ for $0 \leq t < T$, then the sensitivity bound can be defined as the following:*

$$\left\| \frac{\partial h_j^{(T)}}{\partial x_i} \right\|_1 \leq (\alpha\beta)^T (\widehat{A}^T)_{j,i}, \tag{6}$$

*where $\widehat{A}$ denotes the degree-normalized adjacency matrix.*

Now, we show that the sensitivity bound for non-backtracking GNNs can be defined as following. Following Topping et al. (2022), we assume the node features and hidden representations are scalar for better understanding.

**Lemma 1** (**Sensitivity bounds of NBA-GNNs**). *Consider two nodes $i, j \in \mathcal{V}$ with a random walk distance $T$ given a $(T-1)$-layer NBA-GNN as described in Equation (3) and Equation (4). Suppose $\left\| \nabla \phi^{(t)} \right\|_1, \left\| \nabla \sigma \right\|_1 \leq \alpha, \left\| \nabla \psi^{(t)} \right\|_1, \left\| \nabla \rho \right\|_1 \leq \beta$, and $\left\| \nabla \phi \right\|_1 \leq \gamma$ for $0 \leq t < T$. Then the following holds:*

$$\left\| \frac{\partial h_j}{\partial x_i} \right\|_1 \leq (\alpha\beta)^T \gamma (\tilde{C}^\top \widehat{B}^{(T-1)} \tilde{C})_{j,i}.$$

*Proof.* Just a straight calculation using the chain rule is enough to derive the above upper bound.

$$\left\|\frac{\partial h_j}{\partial x_i}\right\|_1 = \left\|\partial_1\sigma\partial_2\rho\left(\sum_{k\to\ell\in N_e^-(j)}\frac{\partial h_{k\to\ell}^{(T-1)}}{\partial x_i}\right) + \partial_1\sigma\partial_3\rho\left(\sum_{k\to\ell\in N_e^+(j)}\frac{\partial h_{k\to\ell}^{(T-1)}}{\partial x_i}\right)\right\|_1$$

$$= \left\|\partial_1\sigma\partial_2\rho\left(\sum_{k\to\ell\in N_e^-(j)}\frac{\partial h_{k\to\ell}^{(T-1)}}{\partial x_i}\right)\right\|_1$$

$$\le \alpha\beta\left(\sum_{k\to\ell\in N_e^-(j)}\left\|\frac{\partial h_{k\to\ell}^{(T-1)}}{\partial x_i}\right\|_1\right), \tag{14}$$

where the bound for the derivatives were $\|\nabla\sigma\|_1 \le \alpha, \|\nabla\rho\|_1 \le \beta$.

Thus considering the message update from Equation (13),

$$\frac{\partial h_{k\to\ell}^{(T-1)}}{\partial x_i} = \partial_1\phi^{(T-2)}\frac{\partial h_{k\to\ell}^{(T-2)}}{\partial x_i} + \partial_2\phi^{(T-2)}\left(\sum_{k'\to\ell'\in N_e^-(k)}\widehat{B}_{k'\to\ell',k\to\ell}\frac{\partial h_{k\to\ell}^{(T-2)}}{\partial x_i}\right)$$

$$= \partial_2\phi^{(T-2)}\left(\sum_{k'\to\ell'\in N_e^-(k)}\widehat{B}_{k'\to\ell',k\to\ell}\frac{\partial h_{k\to\ell}^{(T-2)}}{\partial x_i}\right),$$

since the distance between node $i$ and node $j$ is at least $T$, therefore $\dfrac{\partial h_{k\to\ell}^{(T-2)}}{\partial x_i} = 0$.

Now, let the path from node $i$ to node $j$ as $s(i,j)$, where $s_t$ denotes the $t$-th node in the walk $s$, i.e., $s_0 = i, s_T = j$. Using the bound of the derivative of functions, we can iterate like the following.

$$\left\|\frac{\partial h_{k\to\ell}^{(T-1)}}{\partial x_i}\right\|_1$$

$$\le \alpha\beta\left(\sum_{k'\to\ell'\in\mathcal{N}_e^-(k)}\widehat{B}_{k'\to\ell',k\to l}\left\|\frac{\partial h_{k\to l}^{(T-2)}}{\partial x_i}\right\|_1\right)$$

$$\le (\alpha\beta)^{T-1}\left(\sum_{(s_0,\ldots,s_T)\in s(i,j)}\widehat{B}_{s_0\to s_1,s_1\to s_2}\cdots\widehat{B}_{s_{T-2}\to s_{T-1},s_{T-1}\to s_T}\cdot\left\|\frac{\partial h_{s_0\to s_1}^{(0)}}{\partial x_i}\right\|_1\right)$$

$$= (\alpha\beta)^{T-1}\left(\sum_{(s_0,\ldots,s_T)\in s(i,j)}\prod_{t=1}^{T-1}\widehat{B}_{s_{t-1}\to s_t,s_t\to s_{t+1}}\left\|\frac{\partial h_{s_0\to s_1}^{(0)}}{\partial x_i}\right\|_1\right)$$

$$\le (\alpha\beta)^{T-1}\gamma\left(\sum_{(s_0,\ldots,s_T)\in s(i,j)}\left(\prod_{t=1}^{T-1}\widehat{B}_{s_{t-1}\to s_t,s_t\to s_{t+1}}\right)\right). \tag{15}$$

Substitute the inequality Equation (15) into Equation (14) to get the final result. Then, we can get

$$\left\|\frac{\partial h_j}{\partial x_i}\right\|_1 \le (\alpha\beta)^T\gamma\left(\sum_{(s_0,\ldots,s_T)\in s(i,j)}\left(\prod_{t=1}^{T-1}\widehat{B}_{s_{t-1}\to s_t,s_t\to s_{t+1}}\right)\right)$$

$$= (\alpha\beta)^T\gamma(\tilde{C}^\top\widehat{B}^{T-1}\tilde{C})_{j,i},$$

since paths can be expressed using the power of adjacency-type matrix. □

### C.3 Proof of Proposition 3

We have defined the sensitivity bound for NBA-GNNs. Now, we show that the sensitivity bound of NBA-GNNs are bigger than the sensitivity bound of GNNs.

**Proposition 3.** *Consider an MPNN defined as in Equation* (1) *and a* $(T-1)$*-layer NBA-GNN described by Equation* (3) *and Equation* (4)*. For any pair of nodes* $i, j \in \mathcal{V}$ *with distance* $T$*, the following inequality holds between sensitivity bounds:*

$$(\widehat{A}^T)_{j,i} \leq (\tilde{C}^\top \widehat{B}^{T-1} \tilde{C})_{j,i}.$$

*For* $d$*-regular graphs,* $(\tilde{C}^\top \widehat{B}^{T-1} \tilde{C})_{j,i}$ *decays slower by* $O(d^{-T})$*, while* $(\widehat{A}^T)_{j,i}$ *decays with* $O((d+1)^{-T})$*.*

*Proof.* Identical to the notations above, we denote the list of nodes from node $i$ to node $j$ as path $s(i,j)$ where $s(i,j) = (s_0 = i, s_1, \cdots, s_{T-1}, s_T = j)$.

$$
\begin{aligned}
(\widehat{A}^T)_{j,i} &= \sum_{(s_0,\ldots,s_T) \in s(i,j)} \widehat{A}_{s_0,s_1} \cdots \widehat{A}_{s_{T-1},s_T} \\
&= \sum_{(s_0,\ldots,s_T) \in s(i,j)} (d_i + 1)^{-\frac{1}{2}} \cdot (d_j + 1)^{-\frac{1}{2}} \cdot \prod_{t=1}^{T-1} (d_{s_t} + 1)^{-1}. \\
(\tilde{C}^\top \widehat{B}^{T-1} \tilde{C})_{j,i} &= \sum_{(s_0,\ldots,s_T) \in s(i,j)} d_{s_0}^{-\frac{1}{2}} \cdot \widehat{B}_{s_0 \to s_1, s_1 \to s_2} \cdots \widehat{B}_{s_{T-2} \to s_{T-1}, s_{T-1} \to s_T} \cdot d_{s_T}^{-\frac{1}{2}} \\
&= \sum_{(s_0,\ldots,s_T) \in s(i,j)} d_{s_0}^{-\frac{1}{2}} \cdot d_{s_1}^{-1} \cdots d_{s_{T-1}}^{-1} \cdot d_{s_T}^{-\frac{1}{2}} \\
&= \sum_{(s_0,\ldots,s_T) \in s(i,j)} d_{s_0}^{-\frac{1}{2}} \cdot d_{s_T}^{-\frac{1}{2}} \cdot \prod_{t=1}^{T-1} d_{s_t}^{-1}.
\end{aligned}
$$

Now, for a path $(s_0 = i, \ldots, s_T = j)$,

$$(d_i + 1)^{-\frac{1}{2}} (d_j + 1)^{-\frac{1}{2}} \prod_{t=1}^{T-1} (d_{s_t} + 1)^{-1} \leq d_i^{-\frac{1}{2}} d_j^{-\frac{1}{2}} \prod_{t=1}^{T-1} d_{s_t}^{-1}$$

Each term in $(\tilde{C}^\top \widehat{B}^{T-1} \tilde{C})_{j,i}$ is larger than $(\widehat{A}^T)_{j,i}$, thus $(\tilde{C}^\top \widehat{B}^{T-1} \tilde{C})_{j,i} \geq (\widehat{A}^T)_{j,i}$ is trivial.

For $d$-regular graphs, $d_i = d, \forall i \in \mathcal{V}$. Therefore the sensitivity bound can can be written as following:

$$(\widehat{A}^T)_{j,i} = (d+1)^{-T}, \qquad (\tilde{C}^\top \widehat{B}^{T-1} \tilde{C})_{j,i} = d^{-T}.$$

So $(\tilde{C}^\top \widehat{B}^{T-1} \tilde{C})_{j,i}$ decays slower by $O(d^{-T})$, while $(\widehat{A}^T)_{j,i}$ decays with $O((d+1)^{-T})$.

$\square$

# D    Proofs for Section 4.2

To assess the expressive capabilities, we initially make an assumption about the graphs, considering that it is generated from the Stochastic Block Model (SBM), which is defined as follows:

**Definition 1. *Stochastic Block Model (SBM)* is generated using parameters $(n, K, \alpha, P)$, where $n$ denotes the number of vertices, $K$ is the number of communities, $\alpha = (\alpha_1, ..., \alpha_K)$ represents the probability of each vertex being assigned to communities $\mathcal{V}_1, ..., \mathcal{V}_K$, and $P_{ij}$ denotes the probability of an edge $(v, w) \in \mathcal{E}$ between $v \in \mathcal{V}_i$ and $w \in \mathcal{V}_j$.**

Numerous studies have focused on the problem of achieving exact recovery of communities within the SBM. However, these investigations typically address scenarios in which the average degree is at least on the order of $\Omega(\log n)$ (Abbe, 2017). It is well-established that the information-theoretic limit in such cases can be reached through the utilization of the spectral method, as demonstrated by Yun & Proutière (2016). In contrast, when dealing with a graph characterized by the average degree much smaller, specifically $o(\log n)$, recovery using the graph spectrum becomes a more challenging endeavor. This difficulty arises due to the fact that the $n^{1-o(1)}$ largest eigenvalues and their corresponding eigenvectors are influenced by the high-degree vertices, as discussed in Benaych-Georges et al. (2019).

However, real-world benchmark datasets often fall within the category of very sparse graphs. For example, the citation network dataset discussed in Sen et al. (2008) has an average degree of less than three. In such scenarios, relying solely on an adjacency matrix may not be an efficient approach for uncovering the hidden graph structure. Fortunately, an alternative strategy is available by utilizing a non-backtracking matrix.

## D.1    Proof of Proposition 4

Let's rephrase the formal version of Proposition 4 as follows:

**Proposition 4.**    *Consider a Stochastic Block Model (SBM) with parameters $\left(n, 2, \left(\frac{1}{2}, \frac{1}{2}\right), \left(\frac{a}{n}, \frac{b}{n}\right)\right)$, where $(a + b)$ satisfies the conditions of being at least $\omega(1)$ and $n^{o(1)}$. In such a scenario, the non-backtracking graph neural network can accurately map from graph $\mathcal{G}$ to node labels, with the probability of error decreasing to 0 as $n \to \infty$.*

In Stephan & Massoulié (2022), the authors demonstrate that the non-backtracking matrix exhibits valuable properties when the average degree of vertices in the graph satisfies $\omega(1)$ and $n^{o(1)}$. When $K = 2$, we define a function $\sigma(v)$ for $v \in \mathcal{V}$, such that $\sigma(v) = 1$ if $v$ is in the first class, and $\sigma(v) = -1$ in the second class. Then, they establish the following proposition:

**Proposition 8.** *(Stephan & Massoulié, 2022) Suppose we have a SBM with parameters $\left(n, 2, \left(\frac{1}{2}, \frac{1}{2}\right), \left(\frac{a}{n}, \frac{b}{n}\right)\right)$. In this case, we have two eigenvalues $\mu_1 > \mu_2$ of $ndiag(\alpha)P$, and the eigenvector $\phi_2$ corresponding to $\mu_2$, where $v$-th component is set to $\sigma(v)$. Then, for any $n$ larger than an absolute constant, the eigenvalues $\lambda_1$ and $\lambda_2$ of the non-backtracking matrix $B$ satisfies $|\lambda_i - \mu_i| = o(1)$, and all other eigenvalues of $B$ are confined in a circle with radius $(1 + o(1))\sqrt{\frac{a+b}{2}}$. Also, there exists an eigenvector $\nu_2$ of the non-backtracking matrix $B$ associated with $\lambda_2$ such that*

$$\langle \nu_2, \xi_2 \rangle \geq \sqrt{1 - \frac{8}{(a+b)(a-b)^2}} + o(1) \coloneqq 1 - f(a, b)$$

*where $\xi_2 = \frac{T\Theta\phi_2}{\|T\Theta\phi_2\|}$, $T \in \{0, 1\}^{2m \times n}, T_{ei} = 1\{e_2 = i\}$ and $\Theta \in \{0, 1\}^{n \times 2}, \Theta_{ij} = 1$ if the vertex $i$ is in the $j$-th community, and $0$ otherwise.*

The proposition above highlights that the non-backtracking matrix possesses a spectral separation property, even in the case of very sparse graphs. Moreover, the existence of an eigenvector $\nu_2$ such that $\langle \nu_2, \xi_2 \rangle = 1 - o(1)$ suggests that this eigenvector contains information about the community index of vertices. The foundation for these advantageous properties of the non-backtracking matrix $B$ in sparse scenarios can be attributed to the observation that the matrix $B^k$ shares similarities with the $k$-hop adjacency matrix, while $A^k$ is predominantly influenced by high-degree vertices. Consequently, we can establish the following lemma:

**Lemma 8.** *Suppose we have a SBM with parameters defined in Proposition 8. Then, there exists a function of the eigenvectors of the non-backtracking matrix that can accurately recover the original community index of vertices.*

The proof for Lemma 8 can be found in Appendix D.3 In the following, we will demonstrate that the non-backtracking GNN can compute an approximation to the top $K$ eigenvectors mentioned in Lemma 8. To achieve this, we first require the following lemma:

**Lemma 9.** *Assuming that the non-backtracking matrix $B$ has a set of orthonormal eigenvectors $\nu_i$ with corresponding eigenvalues $\lambda_1 > ... > \lambda_K \geq \lambda_{K+1} \geq ... \geq \lambda_{2m}$, then there exists a sequence of convolutional layers in the non-backtracking GNNs capable of computing the eigenvectors of the non-backtracking matrix.*

For the proof of Lemma 9, please refer to Appendix D.4. With this lemma in mind, we can observe that a sequence of convolutional layers, followed by a multilayer perceptron, can approximate the function outlined in Lemma 8, leveraging the universal approximation theorem. This argument leads to Proposition 4.

## D.2    Proof of Proposition 5

Let's rephrase the formal version of Proposition 5 as follows:

**Proposition 5.** *Consider two graphs, one generated from a SBM ($\mathcal{G}$) with parameters $\left(n, 2, \left(\frac{1}{2}, \frac{1}{2}\right), \left(\frac{a}{n}, \frac{b}{n}\right)\right)$ and the other from an Erdős–Rényi model ($\mathcal{H}$) with parameters $(n, \frac{c}{n})$, for some constants $a, b, c > 1$. When $(a - b)^2 > 2(a + b)$, the non-backtracking graph neural network is capable of distinguishing between $\mathcal{G}$ and $\mathcal{H}$ with probability tending to 1 as $n \to \infty$.*

To establish Proposition 5, we rely on the following Proposition 9 from Bordenave et al. (2015):

**Proposition 9.** *(Bordenave et al., 2015) Suppose we have two graphs $\mathcal{G}$ and $\mathcal{H}$ as defined in the formal statement of Theorem 5. Then, the eigenvalues $\lambda_i(B)$ of the non-backtracking matrix satisfy the following:*

$$\lambda_1(B_{\mathcal{G}}) = \frac{a + b}{2} + o(1), \lambda_2(B_{\mathcal{G}}) = \frac{a - b}{2} + o(1), \quad and \quad |\lambda_k(B_{\mathcal{G}})| \leq \sqrt{\frac{a + b}{2}} + o(1) \quad for \ k > 2$$

$$\lambda_1(B_{\mathcal{H}}) = c + o(1) \quad and \quad |\lambda_2(B_{\mathcal{H}})| \leq \sqrt{c} + o(1)$$

Proposition 9 informs us that by examining the distribution of eigenvalues, we can discern whether a graph originates from the Erdős–Rényi model or the SBM. Leveraging Lemma 9, we can obtain the top two normalized eigenvectors of the non-backtracking matrix using convolutional layers, denoted as $\nu_1$ and $\nu_2$. Applying the non-backtracking convolutional layer to these vectors yields resulting vectors with $\ell_2$-norms corresponding to $\lambda_i(B)$. Consequently, we can distinguish between the two graphs, $\mathcal{G}$ and $\mathcal{H}$, by examining the output of the convolutional layer in the non-backtracking GNN.

## D.3    Proof of Lemma 8

*Proof.* Let us revisit the matrix $T$, defined as $T_{ei} = 1\{e_2 = i\}$, and its pseudo-inverse denoted as $T^+ = D^{-1}T^\top$, where $D$ is a diagonal matrix containing the degrees of vertices on the diagonal. Considering the definition of $\xi_2$ as provided in Proposition 8, we can deduce the label of vertex $v$. Specifically, if $(T^+\xi_2)_v > 0$, it implies that the vertex belongs to the first class; otherwise, it belongs to the other class.

Additionally, we are aware that $\| (T^+\nu_2 - T^+\xi_2)_i \|_2 = O(f(a, b))$ as indicated in Proposition 8, considering the property that the sum of each row of $T^+$ is equal to 1. Consequently, by examining the signs of elements in the vector $T^+\nu_2$, we can classify nodes without encountering any misclassified ones. $\qquad\square$

### D.4 Proof of Lemma 9

*Proof.* Suppose we have $f$ arbitrary vectors $x_1, ..., x_f$ and a matrix $X = [x_1, ..., x_f] \in \mathbb{R}^{2m \times f}$, which has $x_1, ..., x_f$ as columns. Without loss of generality, we assume that $f \leq 2m$ and $\|x_v\|_2 = 1$. Let $x_j = \sum_{i=1}^{2m} c_i^{(j)} \nu_i$ for $1 \leq j \leq f$. We will prove the lemma by showing that if we multiply $X$ by $B$ and repeatedly apply the Gram–Schmidt orthnormalization to the columns of the resulting matrix, the $j$-th column of the resulting matrix converges towards the direction of $\nu_j$. Further, we will show that there exists a series of convolutional layers equivalent to this process.

First, we need to show $B^k x_1$ converges to the direction of $\nu_1$. $B^k x_1$ can be expressed as:

$$B^k x_1 = \lambda_1^k \left( c_1^{(1)} \nu_1 + c_2^{(1)} \left( \frac{\lambda_2}{\lambda_1} \right)^k \nu_2 + ... + + c_{2m}^{(1)} \left( \frac{\lambda_{2m}}{\lambda_1} \right)^k \nu_{2m} \right).$$

Let's fix $\epsilon > 0$ and take $K_0$ such that for all $k \geq K_0$, the inequality $\left( \frac{\lambda_2}{\lambda_1} \right)^k < \frac{\epsilon |c_1^{(1)}|}{4m}$ holds. Furthermore, we define the following for brevity:

$$e_\ell^{(k)} := \begin{cases} \frac{A^k x_1}{\|A^k x_1\|_2} & \text{for } \ell = 1, \\ \frac{u_\ell^{(k)}}{\|u_\ell^{(k)}\|_2} & \text{where } u_\ell^{(k)} = GS(B e_\ell^{(k-1)}) \text{ for } \ell \geq 2 \end{cases}.$$

Then, the distance between $e_1^{(k)}$ and $\text{sign}\left( c_1^{(1)} \right) \nu_1$ is

$$
\left\| e_1^{(k)} - \text{sign}\left( c_1^{(1)} \right) \nu_1 \right\|_2 \overset{(a)}{=} \left\| \frac{c_1^{(1)} \nu_1 + \sum_{i=2}^{2m} \left( \frac{\lambda_i}{\lambda_1} \right)^k c_i^{(1)} \nu_i}{\mathcal{D}} - \text{sign}\left( c_1^{(1)} \right) \nu_1 \right\|_2
$$

$$
\overset{(b)}{<} \left\| \frac{c_1^{(1)} \nu_1 - \text{sign}\left( c_1^{(1)} \right) \mathcal{D} \nu_1}{\mathcal{D}} \right\|_2 + \frac{\epsilon}{2}
$$

$$
= \frac{-|c_1^{(1)}| + \mathcal{D}}{\mathcal{D}} + \frac{\epsilon}{2}
$$

$$
\overset{(c)}{\leq} \frac{\sqrt{\sum_{i=2}^{2m} \left( \frac{\lambda_i}{\lambda_1} \right)^{2k} (c_i^{(1)})^2}}{\mathcal{D}} + \frac{\epsilon}{2}
$$

$$
\overset{(d)}{<} \epsilon,
$$

where (a) stems from defining $\sqrt{(c_1^{(1)})^2 + \sum_{i=2}^{2m} \left( \frac{\lambda_i}{\lambda_1} \right)^{2k} (c_i^{(1)})^2}$ as $\mathcal{D}$, (b) and (d) are obtained from the assumption $\left( \frac{\lambda_2}{\lambda_1} \right)^k < \frac{\epsilon |c_1^{(1)}|}{4m}$, and for (c) we use the inequality $\sqrt{x^2 + y^2} \leq |x| + |y|$.

Moreover, we assume that for all $1 \leq \ell < L$ and any $\epsilon_\ell > 0$, there exists $K_\ell$ such that for all $k \geq K_\ell$, $\left\| e_\ell^{(k)} - \text{sign}\left( c_\ell^{(\ell)} \right) \nu_\ell \right\|_2 < \epsilon_\ell$. To simplify the notations, we define $K_0 = \max_{1 \leq \ell < L} K_\ell$ and $z_\ell^{(k)} = e_\ell^{(k)} - \text{sign}\left( c_\ell^{(\ell)} \right) \nu_\ell$ for all $\ell$. Then, we must show there exists $K_L$ such that for all $k \geq K_L$, $\left\| z_L^{(k)} \right\|_2 < \epsilon_L$. With no loss of generality, let's assume that $\text{sign}\left( c_\ell^{(\ell)} \right) = 1$ for all $\ell$. Furthermore, let us fix $\epsilon_L > 0$ and $\epsilon_\ell = \min(1, \epsilon_0)$

for $1 \le \ell < L$. Then, $u_L^{(k)}$ for $k \ge K_0$ can be written as

$$
\begin{aligned}
u_L^{(k+1)} &= Be_L^{(k)} - \sum_{i<L} \langle e_i^{(k+1)}, Be_L^{(k)} \rangle e_i^{(k+1)} \\
&= Be_L^{(k)} - \sum_{i<L} \left( \langle \nu_i, Be_L^{(k)} \rangle \nu_i + \langle z_i^{(k+1)}, Be_L^{(k)} \rangle \nu_i + \langle \nu_i + z_i^{(k+1)}, Be_L^{(k)} \rangle z_i^{(k+1)} \right) \\
&\overset{(a)}{=} \sum_{i \ge L} \langle \nu_i, Be_L^{(k)} \rangle \nu_i - y_L^{(k)},
\end{aligned}
$$

where (a) comes from the definition $y_L^{(k)} := \sum_{i<L} \langle z_i^{(k+1)}, Be_L^{(k)} \rangle \nu_i + \langle \nu_i + z_i^{(k+1)}, Be_L^{(k)} \rangle z_i^{(k+1)}$.

Additionally, let $d_{\max}$ be the maximum degree of vertices in $\mathcal{G}$. For any vector $x \in \mathbb{R}^{2m}$ with an $\ell^2$-norm less than $\epsilon_x$, the following inequality holds:

$$
\|Bx\|_2 = \sqrt{\sum_i \left( \sum_j b_{ij} x_j \right)^2} < \sqrt{\sum_i \epsilon_x^2 \left( \sum_j b_{ij} \right)^2} \le \epsilon_x d_{\max} \sqrt{2m}. \tag{16}
$$

Thus, using (16), we can deduce that $\left\| y_L^{(k)} \right\|_2 \le 3L\epsilon_0 d_{\max}\sqrt{2m}$.

In contrast, for any integer $p > 0$, $u_L^{(k+p)}$ is

$$
\begin{aligned}
u_L^{(k+p)} &= \sum_{i \ge L} \langle \nu_i, Be_L^{(k+p-1)} \rangle \nu_i - y_L^{(k+p)} \\
&= \frac{1}{\left\| u_L^{(k+p-1)} \right\|_2} \sum_{i \ge L} \left\langle \nu_i, \sum_{j \ge L} \langle \nu_j, Be_L^{(k+p-2)} \rangle \lambda_j \nu_j - B y_L^{(k+p-2)} \right\rangle \nu_i - y_L^{(k+p)} \\
&= \frac{1}{\left\| u_L^{(k+p-1)} \right\|_2} \sum_{i \ge L} \left( \lambda_i \langle \nu_i, Be_L^{(k+p-2)} \rangle - \langle \nu_i, B y_L^{(k+p-2)} \rangle \right) \nu_i - y_L^{(k+p)} \\
&= \frac{1}{\left\| u_L^{(k+p-1)} \right\|_2} \sum_{i \ge L} \langle \nu_i, Be_L^{(k+p-2)} \rangle \lambda_i \nu_i - \frac{1}{\left\| u_L^{(k+p-1)} \right\|_2} \sum_{i \ge L} \langle \nu_i, B y_L^{(k+p-2)} \rangle \nu_i - y_L^{(k+p)} \\
&\;\;\vdots \\
&= \frac{1}{\prod_{j=1}^{p-1} \left\| u_L^{(k+j)} \right\|_2} \sum_{i \ge L} \langle \nu_i, Be_L^{(k)} \rangle \lambda_i^{p-1} \nu_i \\
&\quad - \sum_{j=1}^{p-1} \frac{1}{\left\| u_L^{(k+j)} \right\|_2} \left( \sum_{i \ge L} \langle \nu_i, B y_L^{(k+j-1)} \rangle \lambda_i^{p-j-1} \nu_i \right) - y_L^{(k+p)} \\
&= \frac{1}{\prod_{j=1}^{p-1} \left\| u_L^{(k+j)} \right\|_2} \sum_{i \ge L} \langle \nu_i, Be_L^{(k)} \rangle \lambda_i^{p-1} \nu_i \\
&\quad - \sum_{j=1}^{p-1} \frac{1}{\left\| u_L^{(k+j)} \right\|_2} \left( \sum_{i \ge L} \langle \nu_i, B y_L^{(k+j-1)} \rangle \lambda_i^{p-j-1} \nu_i \right) - y_L^{(k+p)} \\
&\overset{(a)}{=} C_0 \lambda_L^{p-1} \left( \langle \nu_L, Be_L^{(k)} \rangle \nu_L + \sum_{i>L} \langle \nu_i, Be_L^{(k)} \rangle \left( \frac{\lambda_i}{\lambda_L} \right)^{p-1} \nu_i \right) \\
&\quad - \sum_{i \ge L} \sum_{j=1}^{p-1} C_j \left( \langle \nu_i, B y_L^{(k+j-1)} \rangle \lambda_i^{p-j-1} \nu_i \right) - y_L^{(k+p)},
\end{aligned}
$$

where (a) stems from the definitions $C_0 = \frac{1}{\prod_{j=1}^{p-1} \left\| u_L^{(k+j)} \right\|_2}$ and $C_j = \frac{1}{\left\| u_L^{(k+j)} \right\|_2}$.

Now, define $\hat{e}_L^{(k+p)} := \frac{u_L^{(k+p)}}{C_0 \lambda_L^{p-1} \langle \nu_L, Be_L^{(k)} \rangle}$. Then,

$$
\hat{e}_L^{(k+p)} = \nu_L + \sum_{i>L} \frac{\langle \nu_i, Ae_L^{(k)} \rangle}{\langle \nu_L, Be_L^{(k)} \rangle} \left( \frac{\lambda_i}{\lambda_L} \right)^{p-1} \nu_i - \frac{\sum_{i \geq L} \sum_{j=1}^{p-1} C_j \left( \langle \nu_i, By_L^{(k+j-1)} \rangle \lambda_i^{p-j-1} \nu_i \right) - y_L^{(k+p)}}{C_0 \lambda_L^{p-1} \langle \nu_L, Be_L^{(k)} \rangle}. \tag{17}
$$

Take $p_0$ such that $\left( \frac{\lambda_i}{\lambda_L} \right)^{p_0-1} \leq \frac{\epsilon \langle \nu_L, Be_L^{(k)} \rangle}{4(L-N) \langle \nu_i, Be_L^{(k)} \rangle}$, then, the $\ell^2$-norm of the second term of Equation (17) is less than $\epsilon_L/4$.

Meanwhile, the $\ell^2$-norm of the third term in Equation (17) is upper-bounded by

$$
\frac{3L\epsilon_0 d_{\max} \sqrt{2m}}{C_0 \lambda_L^{p-1} \langle \nu_L, Be_L^{(k)} \rangle} \left( Cd_{\max}\sqrt{2m} \sum_{i \geq L} \frac{\lambda_i^{p-2} - 1}{\lambda_i - 1} + 1 \right),
$$

where $C := \max_{j=1}^{p-1} C_j$.

Therefore, if we take $\epsilon_0 < \frac{\epsilon C_0 \lambda_L^{p-1} \langle \nu_L, Be_L^{(k)} \rangle}{12 L d_{\max} \sqrt{2m}} \left( Cd_{\max}\sqrt{2m} \sum_{i \geq L} \frac{\lambda_i^{p-2}-1}{\lambda_i-1} + 1 \right)^{-1}$, we can get $\left\| \hat{e}_L^{(k+p)} - \nu_L \right\|_2 < \epsilon_L/2$ for $p > p_0$. Finally, the upper bound of $\left\| z_L^{(k)} \right\|_2$ is

$$
\begin{aligned}
\left\| z_L^{(k)} \right\|_2 &\leq \left\| e_L^{(k+p)} - \hat{e}_L^{(k+p)} \right\|_2 + \left\| \hat{e}_L^{(k+p)} - \nu_L \right\|_2 \\
&\overset{(a)}{<} \left| \left\| \hat{e}_L^{(k+p)} \right\|_2 - 1 \right| + \frac{\epsilon_L}{2} \\
&\overset{(b)}{\leq} \epsilon_L,
\end{aligned}
$$

where (a) follows from $e_L^{(k+p)} = \frac{\hat{e}_L^{(k+p)}}{\left\| \hat{e}_L^{(k+p)} \right\|_2}$, and (b) is obtained from the inequality $1 - \frac{\epsilon_L}{2} \leq \left\| \hat{e}_L^{(k+p)} \right\|_2 \leq 1 + \frac{\epsilon_L}{2}$. $\qquad\square$

# E    Experiment Details

In this section, we provide details of NBA-GNN implementation and experiments.

## E.1    Implementation

### E.1.1    Message Initialization and Aggregation

For message initialization and final aggregation of messages, we have proposed functions $\phi, \sigma, \rho$. To be specific, the message initialization can be written as follows:

$$h_{i \to j}^{(0)} = \begin{cases} \phi(e_{ij}, x_i, x_j) & (\text{if } e_{ij} \text{ exists}) \\ \phi(x_i, x_j) & (\text{otherwise}) \end{cases}.$$

For our experiments, we use concatenation for $\phi$, weighted sums for $\sigma$, and average for $\rho$.

### E.1.2    Non-backtracking Updates

Here, we provide the details of using the non-backtracking operator in our NBA-GNNS. To begin, let's revisit the non-backtracking matrix $B \in \{0,1\}^{2|\mathcal{E}| \times 2|\mathcal{E}|}$ from Section 4.1 defined as follows:

$$B_{(\ell \to k),(j \to i)} = \begin{cases} 1 & \text{if } k = j, \ell \neq i \\ 0 & \text{otherwise} \end{cases}.$$

Returning to our NBA-GNN, the non-backtracking message passing update for a hidden feature $h_{j \to i}^{(t)}$ at the $(t+1)$-th layer, introduced in Section 3.2, is expressed as follows:

$$h_{j \to i}^{(t+1)} = \phi^{(t)} \left( h_{j \to i}^{(t)}, \left\{ \psi^{(t)} \left( h_{k \to j}^{(t)}, h_{j \to i}^{(t)} \right) : k \in \mathcal{N}(j) \setminus \{i\} \right\} \right), \tag{3}$$

where $\phi^{(t)}$ and $\psi^{(t)}$ are backbone-specific non-linear update and permutation-invariant aggregation functions at the $t$-th layer, respectively. Using the non-backtracking matrix $B$, we can rewrite Equation (3) as following:

$$H^{(t+1)} = \phi^{(t)} \left( H^{(t)}, \psi^{(t)} \left( B^\top H^{(t)}, H^{(t)} \right) \right),$$

where $H^{(t)} \in \mathbb{R}^{2|\mathcal{E}| \times d}$ are edge-wise features, i.e., each row representing $h_{j \to i}^{(t)}$.

### E.1.3    Implementation Example

Now, the NBA-GNN implementation example in Section 3 using GCN (Kipf & Welling, 2017) as a backbone, can be written as the following:

$$h_{j \to i}^{(t+1)} = h_{j \to i}^{(t)} + \frac{1}{|\mathcal{N}(j)| - 1} \mathbf{W}^{(t)} \sum_{k \in \mathcal{N}(j) \setminus \{i\}} h_{k \to j}^{(t)}.$$

Recall the out-degree and in-degree matrices of NBA-GNN, denoted as $D_{out}$ and $D_{in}$, where $(D_{out})_{(j \to i),(j \to i)} = \sum_{\ell \to k} B_{(j \to i),(\ell \to k)}$ and $(D_{in})_{(j \to i),(j \to i)} = \sum_{\ell \to k} B_{(\ell \to k),(j \to i)}$, for each index $j \to i$. We also introduced the normalized non-backtracking matrix augmented with self-loops as $\widehat{B} = (D_{out} + I)^{-\frac{1}{2}}(B + I)(D_{in} + I)^{-\frac{1}{2}}$. In summary, Equation (5) can be expressed as follows:

$$H^{(t+1)} = \widehat{B} H^{(t)} \mathbf{W}^{(t)}.$$

Hence, the message passing update in NBA-GCN is equivalent to the multiplication of non-backtracking operator and edge-wise representations constructed from node-wise features.

### E.2 Long-Range Graph Benchmark

#### E.2.1 Dataset Statistics

From LRGB (Dwivedi et al., 2022), we experiment for 3 tasks: graph classification (`Peptides-func`), graph regression (`Peptides-struct`), and node classification (`PascalVOC-SP`). We provide the dataset statistics in Table 10. Note that for `PascalVOC-SP`, we use SLIC compactness of 30, and edge weights are based only on super-pixels coordinates following the recent work (Gutteridge et al., 2023).

Table 10: Statistics of datasets in LRGB.

| Dataset | Total Graphs | Total Nodes | Avg Nodes | Mean Deg. | Total Edges | Avg Edges | Avg Short. Path. | Avg Diameter | Dataset Splits |
|---|---|---|---|---|---|---|---|---|---|
| PascalVOC-SP | 11,355 | 5,443,545 | 479.40 | 8.00 | 43,548,360 | 3,835.17 | $8.05 \pm 0.18$ | $19.40 \pm 0.65$ | 75/12.5/12.5 |
| Peptides-func | 15,535 | 2,344,859 | 150.94 | 2.04 | 4,773,974 | 307.30 | $20.89 \pm 9.79$ | $56.99 \pm 28.72$ | 70/15/15 |
| Peptides-struct | 15,535 | 2,344,859 | 150.94 | 2.04 | 4,773,974 | 307.30 | $20.89 \pm 9.79$ | $56.99 \pm 28.72$ | 70/15/15 |

#### E.2.2 Experiments Details

All experiment results are averaged over three runs (seed 0∼2) and trained for 300 epochs, with a ∼500k parameter budget. Baseline scores were taken from the LRGB (Dwivedi et al., 2022), Table 1 of DRew (Gutteridge et al., 2023), and papers of each work.

- We use an AdamW optimizer (Loshchilov & Hutter, 2018) with lr decay=0.1 , min lr=1e-5, momentum=0.9, and base learning rate lr=0.001 (0.0005 for `PascalVOC-SP`).

- We use cosine scheduler with reduce factor=0.5 , schedule patience=10 with 50 warm-up.

- Laplacian positional encoding was used with hidden dimension 16 and 2 layers.

- We use the "Atom Encoder", "Bond Encoder" for `Peptides-func`, `Peptides-struct` from based on OGB molecular feature (Hu et al., 2020; 2021), and the "VOCNode Encoder", "VOCEdge Encoder" for `PascalVOC-SP`.

- `PascalVOC-SP` results in Figure 4a all use the same setup described above.

- `Peptides-func` results in Figures 4b and 4c all use the same setup described above.

- GCN results in Figures 4a to 4c use the hyperparameters from Tönshoff et al. (2023).

We searched the following range of hyperparameters, and reported the best in Table 11.

- We searched layers 6 to 12 for `PascalVOC-SP` 2, layers 5 to 20 for `Peptides-func` and `Peptides-struct`.

- The hidden dimension was chosen by the maximum number in the parameter budget.

- Dropout was searched from 0.0∼0.8 for `PascalVOC-SP` in steps of 0.1, and 0.1∼0.4 in steps of 0.1 for `Peptides-func` and `Peptides-struct`.

- We used the batch size of 30 for `PascalVOC-SP` on GPU memory, and 200 for `Peptides-func` and `Peptides-struct`.

Table 11: Best hyperparameters for each NBA-GNN and dataset in LRGB.

| Model | Dataset | # Param. | # Layers | hidden dim. | dropout | Batch size | # epochs |
|---|---|---|---|---|---|---|---|
| NBA-GCN | PascalVOC-SP | 472k | 12 | 180 | 0.7 | 30 | 200 |
| | Peptides-func | 510k | 10 | 186 | 0.1 | 200 | 300 |
| | Peptides-struct | 505k | 20 | 144 | 0.1 | 200 | 300 |
| NBA-GIN | PascalVOC-SP | 472k | 12 | 180 | 0.7 | 30 | 200 |
| | Peptides-func | 502k | 10 | 144 | 0.1 | 200 | 300 |
| | Peptides-struct | 503k | 10 | 144 | 0.1 | 200 | 300 |
| NBA-GatedGCN | PascalVOC-SP | 486k | 10 | 96 | 0.25 | 30 | 200 |
| | Peptides-func | 511k | 10 | 96 | 0.1 | 200 | 300 |
| | Peptides-struct | 511k | 8 | 108 | 0.1 | 200 | 300 |

### E.3 Transductive Node Classification

#### E.3.1 Dataset statistics

We conducted experiments involving three citation networks (Cora, CiteSeer, and Pubmed in Sen et al. (2008)), and three heterophilic datasets (Texas, Wisconsin, and Cornell in Pei et al. (2019)), focusing on transductive node classification. Our reported results in Table 3 are the averages obtained from 10 different seed runs to ensure robustness and reliability.

For the citation networks, we employed the dataset splitting procedure outlined in Yang et al. (2016). In contrast, for the heterophilic datasets, we randomly divided the nodes of each class into training (60%), validation (20%), and testing (20%) sets. We provide more details of dataset statistics in Table 12.

#### E.3.2 Experiment Details

The training duration spanned 1,000 epochs for citation networks and 100 epochs for heterophilic datasets. Following training, we selected the best epoch based on validation accuracy for evaluation on the test dataset. We used the AdamW optimizer (Loshchilov & Hutter, 2018) with a learning rate of 3e-5. The model's hidden dimension and dropout ratio were set to 512 and 0.2, respectively, consistent across all datasets, after fine-tuning these hyperparameters on the Cora dataset. Additionally, we conducted optimization for the number of convolutional layers within the set $\{1, 2, 3, 4, 5\}$. The results revealed that the optimal number of layers is typically three for most of the models and datasets. However, there are exceptions, such as CiteSeer-GatedGCN, PubMed-{GraphSAGE, GraphSAGE+NBA+PE}, Wisconsin-{GraphSAGE+NBA, GraphSAGE+NBA+PE} and Cornell-{GraphSAGE, GraphSAGE+NBA, GraphSAGE+NBA+PE}, where the optimal number of layers is found to be four. Furthermore, for Cora-{GraphSAGE+NBA+PE, GAT+NBA}, CiteSeer-GraphSAGE+NBA, the optimal number of layers is determined to be five.

Table 12: Statistics of the datasets for the transductive node classification task

| Dataset | Total Graphs | Num Nodes | Num Edges | Dim Features | Num Classes |
|---|---|---|---|---|---|
| Cora | 1 | 2,708 | 10,556 | 1,433 | 7 |
| CiteSeer | 1 | 3,327 | 9,104 | 3,703 | 6 |
| PubMed | 1 | 19,717 | 88,648 | 500 | 3 |
| Texas | 1 | 183 | 309 | 1,703 | 5 |
| Wisconsin | 1 | 251 | 499 | 1,703 | 5 |
| Cornell | 1 | 183 | 295 | 1,703 | 5 |

### E.3.3 Baseline implementation

In Section 5.3, we compared NBA-GNNs with several baselines to verify the effectiveness of non-backtracking updates. Three baselines that update edge features were considered: GatedGCN (Bresson & Laurent, 2018), EGNN (Gong & Cheng, 2019), and CensNet (Jiang et al., 2020). Though these baselines update edge features, the six datasets used for transductive node classification do not have initial edge features. Based on each paper and its code, we used the pairwise cosine similarities between corresponding node features as edge features for CensNet, and encoded each directed edge into a vector $\mathbf{v} \in \{0, 1\}^3$ for EGNN. Additionally, we utilize the attention-based EGNN model, referred to as EGNN(A) in the original paper.

