# OpenReview forum: "Non-backtracking Graph Neural Networks"
_TMLR — Accepted by TMLR_

### Review · Reviewer_P53w · 2024-07-02

**Summary Of Contributions:**

This paper proposes a non-backtracking (NBA) technique for message passing in graph neural networks.
It considers updating edge features and building non-redundant computation graphs for the message passing neural networks (MPNNs).

**Audience:**

Yes

**Claims And Evidence:**

No

**Requested Changes:**

The following comments or suggestions need to be considered.

1. The evaluation is expected to be enhanced. Several baseline methods have been considered, including GCN, GINE, and GateGCN, to show the effectiveness of
the proposed technique. These models are standard, but more and recent models are  expected to be considered. Moreover, these baseline models appear to be mainly
based on updating of node features. Because the proposed NBA-MPNN is mainly based on updating of edge features, which might be complementary to the updating of node features, it is unclear if the experimental improvement is mainly due to the NBA, or due to the message passing over edge features. I would suggest the authors to consider also the following models as baselines, which are mainly based on or consist of updating of (multi-dimensional) edge features:

- “Co-embedding of nodes and edges with graph neural networks,” IEEE Transactions on Pattern Analysis and Machine Intelligence, 2020.
- "Exploiting edge features for graph neural networks,” in 2019 IEEE/CVF Conference on Computer Vision and Pattern Recognition (CVPR), 2019.
- "Graph transformer networks,” in Advances in Neural Information Processing Systems, 2019.
- "Graph-mamba: Towards long-range graph sequence modeling with selective state spaces," arXiv preprint, 2024.


2. Please clearly state what insights or what impacts the theoretical results give/have on the algorithm or experiments.

3. In the experimental part, the partitioning of the LRBG such as Peptides-func, -struct, and Pascal-VOC is specified. Please specify because the performance is impacted by the partitioning ratios.

Based on the review results, I would recommend review again after major revision.

**Strengths And Weaknesses:**

Strengths:
- The idea appears effective, albeit simple. It seems to be an interesting generalization
of the non-backtracking random walks to message passing.
- Theoretical results are derived.
- Some experimental results show the effectiveness of the proposed technique to some extent.


Weaknesses:
- Experimental evaluations use limited baselines and datasets.
- Some experimental details are missing.
- The insights from theoretical results or their significance on the algorithms or
parameter configurations or tuning are not so clear. Theoretical results and
practical side, e.g., the use of the algorithm on experiments, seem disconnected.

---

> ### Author Response · Authors · 2024-08-13
> **Response to Reviewer P53w**
>
> Dear Reviewer P53w,
>
> We thank you for your interest in our work and the valuable feedback you have provided. Below, we have addressed the weaknesses and requested changes one by one. We have highlighted the changes in blue in the updated manuscript.
>
> ---
> **W1/C1. Lack of recent models [1] and edge feature updating models [2, 3].**
>
> Note that Table 2 already includes recent baselines such as PathNN [4], CIN++ [5], Exphormer [6], Graph MLP-Mixer/ViT [7] and DRew [8], where CIN++ and Graph MLP-Mixer/ViT are edge feature updating models. Nevertheless, to resolve your concern, we have added Graph-mamba [1] (a recent model), respectively. Notably, NBA-GNN still outperforms most of these recent baselines, showing the effectiveness of non-backtracking updates.
>
> We also added EGNN [2] and CensNet [3] (both edge feature updating models) in Table 4 (please refer to Appendix E.3.3 for experiment details). One can observe that NBA-GNN still demonstrates improvements compared to these new baselines. We have decided to exclude the Graph transformer network [9] from the comparison, as it is specifically designed for heterogeneous graph datasets with multiple types of nodes and edges.
>
> ---
> **C1. Unclear origin of experimental improvements in LRGB (whether due to the NBA or message passing over edge features)**
>
> In our ablation study in Section 5.3, we showed that experimental improvements mainly come from non-backtracking updates. To be specific, we have tested $\textit{backtracking}$ GNNs (BA-GNN) which remove the non-backtracking condition, i.e., replacing ${k \in \mathcal{N}(j) \setminus \{i\}}$ by ${k \in \mathcal{N}(j)}$ in equation (3). As shown in Figures 4a and 4b, NBA-GNN consistently outperforms BA-GNN regardless of the number of layers, confirming that experimental improvement truly comes from non-backtracking updates.
>
> ---
> **W2/C3. Missing experimental details**
>
> We have followed the standard dataset split of the long-range graph benchmark [10], and added the details in Table 9 of Appendix E (experiment details). The train-valid-test split of peptides-func and peptides-struct is 70/15/15, and the split of Pascal-VOC is 75/12.5/12.5. For citation networks, we followed the dataset-splitting procedure from the original paper [11], as noted in Appendix E.3. Also, heterophilic datasets were split using a 60/20/20 train-valid-test ratio.
>
> ---
> **W3/C2. Insights and impacts of theoretical results on algorithm or experiments**
>
> We have clarified the insights and impacts of our theoretical results in Sections 4.1 and 4.2 (Note that Theorem 1 has been modified to Proposition 3) . In Section 4.1 and Appendix C, we demonstrated that the sensitivity bound in NBA-GNN is larger compared to conventional GNNs. This finding allows us to compare the degree of over-squashing across message-passing algorithms aligning with different random walks, which, to the best of our knowledge, is the first. Furthermore, since NBA-GNN has a larger sensitivity bound, it is feasible to stack more backbone GNN layers in practice without suffering from over-squashing, as shown in Figure 4b.
>
> In Section 4.2, we demonstrate the expressive capabilities of NBA-GNNs (Note that Theorem 2 and 3 have been modified to Proposition 4 and 5). Proposition 4 suggests that even when a graph is too sparse to extract valuable node-wise information using a GNN with the adjacency matrix, NBA-GNN can still successfully classify the nodes. Similarly, Proposition 5 establishes this for graph classification. The experimental results in Table 3 show the non-backtracking update improves the performance of all GNN variants for node classification.
>
> ---
> **References**
>
> [1] Graph-Mamba: Towards long-range graph sequence modeling with selective state spaces (Wang et al., arXiv 2024)
>
> [2] Exploiting edge feature for graph neural networks (CVPR 2019)
>
> [3] Co-embedding of nodes and edges with graph neural networks (TPAMI 2020)
>
> [4] Path Neural Networks: Expressive and accurate graph neural networks (ICML 2023)
>
> [5] CIN++: Enhancing topological message passing (ICML 2023)
>
> [6] Exphormer: Sparse Transformer for graphs (ICML 2023)
>
> [7] A generalization of ViT/MLP-Mixer to graphs (ICML 2023)
>
> [8] DRew: Dynamically rewired message passing with delay (ICML 2023)
>
> [9] Graph transform networks (NIPS 2019)
>
> [10] Long-range graph benchmark (NIPS 2023)
>
> [11] Revisiting semi-supervised learning with graph embeddings (ICML 2016)

---

> > ### Comment · Reviewer_P53w · 2024-08-29
> >
> > Thanks for your response. I am okay with the revision and responses.

---

### Review · Reviewer_pxoD · 2024-07-12

**Summary Of Contributions:**

This submission analyzes Non-Backtracking Graph Neural Networks (NBA-GNN) in the line of Chen al. (2019) (but refined). These networks address the redundancy in message passing of typical MPGNNs by preventing messages from backtracking to previously visited nodes. This approach reduces redundancy, alleviates the over-squashing phenomenon, and enhances the expressive power of GNNs, particularly in distinguishing structures within sparse stochastic block models. Empirical validations demonstrate that NBA-GNNs moderatly outperform conventional GNNs in various graph-related tasks.

Despite several flaws that I describe below, I believe that the paper is a good fit for TMLR. The paper is well-written and structured, and the idea of NBA-GNN is interesting.

**Audience:**

Yes

**Claims And Evidence:**

No

**Requested Changes:**

p6:  Can you confirm that |.| is the L2-norm and ||.|| is the spectral (or Frobenius I have a doubt) norm? If I am not mistaken the Jacobian considered is of size $p_T \times p_0$ where $p_i$ are the sizes of the features.

p6: You do not define what is a sensitivity bound. I guess this is the r.h.s of the equation in Lemma 1.

p28: third line of eq 14. that's a lot of dots!

p31: the notation $f(\leq 2m)$ is a bit confusing.

p32: You simplify a lot the big block of equation by naming the numerator.

all documents: please check typography (display equation environment should be ponctuated for instance).

**Strengths And Weaknesses:**

### Strengths
1. The paper is well-written and clearly structured.

2. This is a welcome take on the over-squashing problem in GNNs. I believe that imposing non-backtracking constraints is a good idea, be it using the architecture proposed in the paper, or thinking of alternative solutions.

3. Appendix A is a nice addition to the paper, I would definitely move it inside the main text! You should take advantage of the fact that it is a journal paper ...

### Weaknesses

1. The novelty of the method is not that obvious with respect to previous works such as Chen et al. (2019). Still, the paper provides a refined version of the idea, allowing true non-backtracking flow, and I believe this is sufficiently "novel" in this sense for TMLR.

2. The lack of discussion (except for two sentences) of the expresivity of NBA-GNN w.r.t $k$-WL tests undermines the expressivity claims of the paper.

3. The authors should discuss if the assumptions in Lemma 1 are "reasonable" or "natural".

4. The statement of Theorem 1 is too informal. Also calling "theorem" the content in Theorem 2 & 3 seems a bit of an overstatement. They are immediate consequences of Lemma 9 if I read correctly the appendix.

5. I think a discussion is missing after Theorem 1. Can we believe this bounds to be optimal? Does this decay can be observed on $d$-regular graph? What happens for almost regular graph?

6. The organization of the proof section is not optimal. There is some forward reference to results proved afterwards (e.g. Lemma 9 depends on Lemma 10)

---

> ### Author Response · Authors · 2024-08-13
> **Response to Reviewer pxoD (1/2)**
>
> Dear Reviewer pxoD,
>
> We are grateful for your valuable time and helpful comments on our paper. As you mentioned, our work introduces $\textit{true non-backtracking flows}$ unlike previous approaches and revealed that it theoretically alleviates over-squashing. In response to your suggestion, we have moved the complexity analysis from Appendix A.1 to Section 5 (experiment). Below, we address the weaknesses and the requested changes one by one.
>
> ---
> **W2. Lack of discussion of the expressivity of NBA-GNN with respect to $k$-WL**
>
> We have added more details in Section 4.2 regarding the discussion on the expressivity of NBA-GNN with the $k$-WL test, summarized as follows. Recent works have found that (i) the $k$-WL test is inadequate for measuring the expressive power in node classification tasks, and (ii) it has a wide performance gap between the 1-WL (which is equivalent to 2-WL [1]) and 3-WL tests [2], making it difficult to compare the proposed algorithms within this range. Therefore, alternative metrics for evaluating the expressivity of GNNs have been suggested [2,3,4]. Building on previous studies [5,6,7], we have focused on the $\textit{spectral analysis on the SBM}$ to analyze the theoretical performance of NBA-GNNs.
>
> ---
> **W3/C2. Lemma 1: Insufficient information and discussion on assumptions**
>
> We have relocated the definition of the sensitivity bound [8] from Appendix C.2 to Section 4.1, and added an explanation for clarity. Furthermore, we have added the details on the assumptions in Lemma 1, summarized as follows. For the assumptions on the nabla bounds in Lemma 1, it is quite natural considering the Lipschitz constant of the nonlinearity function and the maximum entry value across all weight matrices [9]. Additionally, these assumptions are consistent with those in previous work [8].
>
> ---
> **W4/W5. Theorem 1: Informal statement. Consideration of optimal bounds.**
>
> We have rewritten Theorem 1 as Proposition 3, and refined its statement for formality. In short, we have shown that the sensitivity bounds of NBA-GNNs are larger than that of conventional GNNs, based on comparing the number of non-backtracking and simple walks. For the optimality of the bound, we would like to note that achieving tightness of the sensitivity bounds requires unrealistic strong assumptions, e.g., the existence of parameters that set the sensitivity term to zero. Nevertheless, our main contribution lies in being the first to compare the degree of over-squashing between GNNs aligning with different random walks, which modifies the graph topology [9].
>
> ---
> **W5. Missing discussion for Theorem 1, decay in $d$-regular graphs and almost regular graphs.**
>
> We have added detailed discussion and decays of $d$-regular graph after Theorem 1 (modified to Proposition 3). For $d$-regular graphs, the sensitivity bound of NBA-GNN decays more slowly by ${O(d^{-T})}$ compared to that of conventional GNNs, ${O((d+1)^{-T})}$. Intuitively, one can consider multiplying the power of a non-backtracking matrix or adjacency matrix with a one-hot vector, where all entries of the resulting vector by the non-backtracking matrix will be larger than using the adjacency matrix. In the case of almost regular graphs [10,11], i.e., graphs with every vertex degree inside a small bound and not being a $d$-regular graph, we assume a similar decay pattern will be observed but we leave the detailed proof of specific graph types for future works.
>
> ---
> **W4. Overstatement of Theorem 2 & 3, immediate consequence of Lemma 9**
>
> Thank you for your feedback. We acknowledge your concern that Theorem 2 and 3 may be overstated as they are immediate consequences of Lemma 9. In response, we have rewritten them as Proposition 4 and 5 to reflect their derivation from Lemma 9 more accurately.
>
> ---
> **W6,C3,C4,C5. Organization of the proof section, typography, and minor errors**
>
> We thank the reviewer for pointing this out. Below, we list the proof sections we have rearranged in the appendix, with changes highlighted in blue in the paper.
>
> - Appendix B.2, proposition 6
>
> - Appendix B.3, proposition 7
>
> - Appendix C.1, proposition 3, lemma 1
>
> - Appendix D.4, lemma 9
>
> We also appreciate the reviewer for pointing out typos, punctuation errors, and notation issues. We have made all the necessary corrections throughout the paper and appendix listed below, with changes highlighted in blue.
> Overall display equation punctuations
>
> - p28: too many dots
>
> - p31: notation for the number of eigenvectors
>
> - p32: simplifying the equation by naming the denominator

---

> ### Author Response · Authors · 2024-08-13
> **Response to Reviewer pxoD (2/2)**
>
> **C1. Notation of norms**
>
> The notation $\vert\cdot\vert$ in the assumptions and partial derivatives of lemma 1 represents the L1-norm. This is because we also assumed that the node features and hidden representations to be scalar for better understanding, and the proof can be easily extended to higher-dimension cases without much modification . Based on the standard notation guide of TMLR, we have updated all L1–norm notations as $\Vert\cdot\Vert_{1}$, L2-norm notations as $\Vert\cdot\Vert_{2}$ or $\Vert\cdot\Vert$. For the size of the Jacobian, you are correct: we consider the L1-norm of a matrix size $p_T \times p_0$, where $p_i$ is the feature size at layer $i$.
>
> ---
> **Reference**
>
> [1] A short tutorial on the Weisfeiler-Lehman test and its variants (ICASSP 2021)
>
> [2] $\mathcal{N}$-WL: A new hierarchy of expressivity for graph neural networks (ICLR 2023)
>
> [3] Rethinking the expressive power of GNNs via graph biconnectivity (ICLR 2023)
>
> [4] Beyond Weisfeiler-Lehman: A quantitative framework for gnn expressiveness (ICLR 2024)
>
> [5] Graph Neural Networks Exponentially Lose Expressive Power for Node Classification (ICLR 2020)
>
> [6] Graph attention retrospective (JMLR 2023)
>
> [7] Effects of graph convolutions in multi-layer networks (ICLR 2023)
>
> [8] Understanding over-squashing and bottlenecks on graph via curvature (ICLR 2022)
>
> [9] On over-squashing in message passing neural networks: The impact of width, depth, and topology (ICML 2023)
>
> [10] Regular subgraphs of almost regular graphs(Journal of combinatorial theory series B  1984)
>
> [11] The reflexive edge strength on some ​​almost regular graph (Heliyon 2021)

---

### Review · Reviewer_ixiA · 2024-07-30

**Summary Of Contributions:**

This paper introduces the non-backtracking graph neural network (NBA-GNN) to address the redundancy issue in message-passing updates of traditional GNNs. The authors propose a method that prevents messages from revisiting previously visited nodes, thereby reducing over-squashing and improving performance on long-range graph benchmarks and node classification tasks.

**Audience:**

Yes

**Broader Impact Concerns:**

NA.

**Claims And Evidence:**

Yes

**Requested Changes:**

- The explanation for “Step 1” in Figure 1 is missing, making it challenging for readers to grasp the distinction between “Step 1” and “Step 2”. It would be better to add more explanations.

- Consider including an experiment or case study to demonstrate how the proposed update can effectively reduce redundancy, if feasible.

- It would be better to standardize the baselines for the two tasks in experiments, if feasible. Otherwise, please clarify the criteria used for selecting these baselines.

**Strengths And Weaknesses:**

Pros:

- Implementing non-backtracking updates offers a novel solution to addressing the redundancy issue in GNNs.

- The paper conducts a comprehensive theoretical analysis, demonstrating that NBA-GNN enhances the upper bound for sensitivity-based measures of GNN over-squashing and can identify the underlying structure of SBMs, even in very sparse graphs.

- The experiments on two benchmarks show that the proposed GNN can improve existing GNN baselines.

Cons:

- The study aims to address the redundancy problem in GNNs. However, the experiments lack direct comparisons or specific case studies demonstrating that the proposed non-backtracking updates effectively reduce redundancy. In my view, the discussions in Appendix A do not adequately support this claim.

- The baselines presented in Tables 2 and 3 appear outdated. The criteria for selecting these baselines are unclear, which weakens the soundness of the work.

---

> ### Author Response · Authors · 2024-08-13
> **Response to Reviewer ixiA**
>
> Dear Reviewer ixiA,
>
> We thank you for your valuable time for reviewing our paper. Below, we have listed your concerns one by one and hope it would resolve the weaknesses and requested changes.
>
> ---
> **W1/C2. Experiments lacking direct comparison or case studies to demonstrate non-backtracking updates reduce redundancy**
>
> In section 5.3, we conducted ablation studies to show that non-backtracking reduces redundancy and leads to performance improvement, comparing NBA-GNN with $\textit{backtracking}$ GNN, i.e., BA-GNN. BA-GNN removes the non-backtracking condition, replacing ${k \in \mathcal{N}(j) \setminus \{i\}}$ by ${k \in \mathcal{N}(j)}$ in Equation (3). In Figures 4a and 4b, NBA-GNN consistently outperforms BA-GNN regardless of the number of layers, confirming that non-backtracking updates have reduced redundancy and thereby lead to performance improvements.
>
> Furthermore, in section 5.3 in the updated manuscript, we also compare NBA-GNN with edge-feature updating models [1,2] as reviewer P53w suggested. From Table 4, one can see that NBA-GNN outperforms other edge feature updating models, further showing the effectiveness of non-backtracking.
>
> ---
> **W2/C3. Outdated baselines and unclear criteria of baseline selection**
>
> For the LRGB experiments in Table 2, we show the competitive performance of NBA-GNN over baselines. The baseline selection criteria is the recent works targeting the long-range graph benchmark (LRGB) [3]: PathNN [4], CIN++ [5], Exphormer [6], Graph MLP-Mixer/ViT [7], and DRew [8]. Based on reviewer P53w’s suggestions, we have also added Graph-Mamba [9] to the table. For the backbone network, we used conventional GNNs based on prior work, DRew [8]. Nevertheless, one can observe that NBA-GNN still demonstrates superior performance compared to new baselines.
>
> For transductive node classification in Table 3, we show the performance improvement of NBA variants, similar to Table 1. Nevertheless, for further investigation, we have additionally tested the following models in Table 4, EGNN [1] and CensNet [2], GNNs that update edge features as reviewer P53w suggested (please refer to Appendix E.3.3 for experiment details). Notably, NBA-GNN also outperforms these models on most datasets, showing that performance improvements originate from non-backtracking.
>
> ---
> **C1. Missing explanation for step 2 in Figure 1**
>
> Thank you for pointing this out. We have added more explanation for step 1 and 2 in Figure1, highlighting the difference between simple updates and non-backtracking updates considering the message flow.
>
> ---
> **Reference**
>
> [1] Exploiting edge feature for graph neural networks (CVPR 2019)
>
> [2] Co-embedding of nodes and edges with graph neural networks (TPAMI 2020)
>
> [3] Long-range graph benchmark (NIPS 2023)
>
> [4] Path Neural Networks: Expressive and accurate graph neural networks (ICML 2023)
>
> [5] CIN++: Enhancing topological message passing (ICML 2023)
>
> [6] Exphormer: Sparse Transformers for Graphs (ICML 2023)
>
> [7] A generalization of ViT/MLP-Mixer to graphs (ICML 2023)
>
> [8] DRew: Dynamically rewired message passing with delay (ICML 2023)
>
> [9] Graph-mamba: Towards long-range graph sequence modeling with selective states spates (Wang et al., arXiv 2024)

---

> > ### Comment · Reviewer_ixiA · 2024-08-29
> >
> > Thanks for your response.

---

### Comment · Reviewer_P53w · 2024-08-29

Thanks for your responses. I am okay with the revision and responses.

---

### Decision · Action_Editor_Ekvd · 2024-09-05

**Recommendation:** Accept as is

**Comment:**

The paper was appreciated by the reviewers that had many questions and suggestions. Those suggestions were very well taken into account by the authors in the revision which was appreciated by the reviewers. All reviewers believe that the paper should be accepted and I concur.

**Audience:**

This paper about GNN, their limits (oversmoothing) and how to attenuate them is definitely of interest to the community.

**Claims And Evidence:**

All claims are supported by clear evidence either theoretical proof or empirical experiments.